# Direct Flow Q-Learning

**Shicheng Cao**[1]  **Jingrui Jia**[1]  **Wenyu Li**[1]  **Feng Duan**[1,2]  **Tao Zhang**[3]  **Shengbo Li**[4]

## Abstract

Flow Matching shows great promise in offline reinforcement learning (RL), yet optimizing these iterative policies via Backpropagation Through Time (BPTT) is unstable. While prevailing paradigms circumvent this by distilling multi-step flows into single-step approximations, such methods may limit the benefits of iterative refinement. To avoid these sacrifices, we propose Direct Flow Q-Learning (DFQL), a streamlined framework that attains superior results by optimizing flow matching policies without BPTT or distillation. DFQL derives a surrogate objective that directly injects terminal Q-value gradients as a guidance term into each step velocity field, ensuring stable optimization while preserving iterative expressive capacity. Across 73 challenging tasks in OGBench and D4RL, DFQL achieves state-of-the-art results. Additionally, DFQL extends seamlessly to the offline-to-online setting, delivering substantial performance gains without further modification. Code is available at https://github.com/2012060/DFQL.

## 1. Introduction

Offline reinforcement learning (RL) aims to learn a policy from a fixed dataset of previously collected interactions, without additional environment interaction during training (Levine et al., 2020; Prudencio et al., 2024), which can reduce the cost and risk of online data collection, and provide a strong initialization for subsequent online interaction and fine-tuning (Nakamoto et al., 2023). This setting is particularly critical in domains such as autonomous driving (Guan et al., 2024) and robotic manipulation (Intelligence et al.,

[1]College of Artificial Intelligence, Nankai University, Tianjin, China [2]Tianjin Key Laboratory of Interventional Brain-Computer Interface and Intelligent Rehabilitation, Nankai University, Tianjin, China [3]SunRisingAI Ltd., Beijing, China [4]School of Vehicle and Mobility, Tsinghua University, Beijing, China. Correspondence to: Wenyu Li <liwenyu@nankai.edu.cn>.

*Proceedings of the 43$^{rd}$ International Conference on Machine Learning*, Seoul, South Korea. PMLR 306, 2026. Copyright 2026 by the author(s).

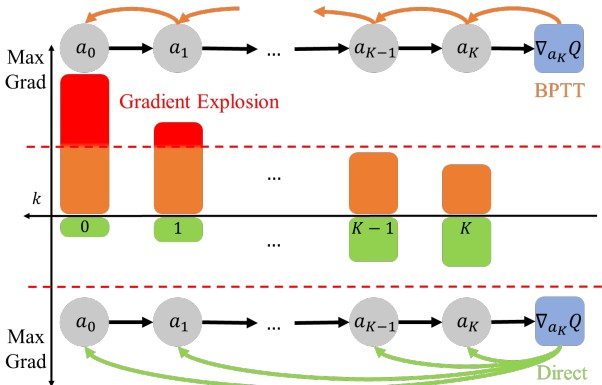

*Figure 1.* Flow-matching-based offline RL suffers from BPTT instability, as backpropagating terminal critic signals step-by-step through multiple discrete steps may lead to exploding gradients (orange). Hence, we propose a surrogate objective that directly injects weighted terminal Q-gradients into the velocity field at each step, bypassing this unstable recursive backpropagation path to achieve more stable training (green).

2025). Consequently, offline RL is inherently constrained by the behavior distribution reflected in the logged dataset (Koppel et al., 2024). Unlike online RL, deviations from this distribution cannot be corrected through interaction, and aggressively optimizing for a single unconstrained optimum may lead to unreliable estimates due to out-of-distribution (OOD) actions (Chen et al., 2025a). As datasets grow in scale and behavioral diversity, the induced action distributions often become highly multimodal, making the representational capacity of the policy class a critical factor for effective offline RL (Qing et al., 2024; Venkatraman et al., 2024). Motivated by this challenge, recent advances in flow matching networks, known for their strong multimodal modeling capabilities, have provided a promising foundation for expressive policy learning (Song et al., 2021; Liu et al., 2023; Lipman et al., 2023). A growing body of work has begun to integrate flow-matching-based models into offline RL, reporting consistent performance gains across diverse benchmarks (Zhang et al., 2025a; Nguyen & Yoo, 2026).

However, the integration of flow matching policies with RL objectives presents a fundamental optimization challenge. While flow matching excels at density estimation by supervising the model in the velocity (vector-field) space, RL objectives are inherently defined in the terminal action

space (Ben-Hamu et al., 2024). This discrepancy forces a difficult choice in policy extraction. On one hand, gradient-free approaches, such as value-weighted behavior cloning or critic-guided sampling (Peters & Schaal, 2007; Chen et al., 2023), avoid the complexities of differentiating through the flow steps but often rely on indirect guidance, which can limit the exploitation of the critic's fine-grained landscape in high-dimensional spaces (Wang et al., 2023). On the other hand, seeking to optimize the policy via end-to-end differentiation of the multi-step generation process (i.e., BPTT) is unstable, often suffering from exploding gradients (Ren et al., 2025). Consequently, achieving a stable yet direct coupling between the iterative vector field and the RL objective remains an open challenge.

To circumvent the instability of BPTT, existing literature primarily follows two distinct directions. The prevailing paradigm involves policy distillation, which compresses the multi-step flow into a single-step approximation for standard RL optimization (Park et al., 2025b; Tiofack et al., 2025). Although achieving high efficiency, this simplification imposes a capacity bottleneck, as the reduction of multimodal trajectories into a simpler form constrains the iterative refinement process critical for complex task performance (Wang et al., 2026). Alternatively, another line of work seeks to stabilize BPTT directly by employing specialized architectures, such as gating mechanisms or sequential modeling, to facilitate gradient flow through the multi-step integration path (Zhang et al., 2025c). However, these structural modifications often entail implementation overhead and impose rigid architectural requirements, thereby limiting their versatility across diverse neural network designs. This raises a critical question: Can we leverage terminal Q-value signals to optimize flow matching policies while preserving their multi-step iterative structure, without imposing additional constraints on the underlying network architecture?

In this work, we exploit the structural properties of flow matching networks to propose Direct Flow Q-Learning (DFQL), a framework that injects RL objectives directly into the velocity space. Instead of backpropagating terminal Q signals through the multi-step integration path, DFQL distributes the Q gradients across all generation steps as step-wise supervision (Figure 1). We provide a theoretical analysis establishing our method as a stable approximation to BPTT. By omitting components prone to triggering gradient explosion, we effectively trade precision for enhanced numerical robustness throughout the optimization process. From another perspective, our approach can be viewed as moving inference-time guidance (Jang et al., 2025) into the training objective. This ensures the model learns a vector field that is already "value-aware" without needing extra guidance at test time. By avoiding recursive gradients, DFQL ensures stable, architecture-agnostic optimization while preserving the benefits of iterative refinement. Simi-

lar to FQL, DFQL is simple to implement within standard Actor-Critic pipelines, yet consistently outperforms baselines across 73 OGBench (Park et al., 2025a) and D4RL (Fu et al., 2020) tasks. These results bridge the gap between training stability and policy expressivity, while naturally supporting further gains via online fine-tuning.

## 2. Preliminaries

### 2.1. Offline RL

In this work, we model the environment as a discounted Markov decision process (MDP) $\mathcal{M} = (\mathcal{S}, \mathcal{A}, p, r, \rho, \gamma)$, where $\mathcal{S}$ and $\mathcal{A} \subseteq \mathbb{R}^d$ denote the state and action spaces, $r$ is the reward function, $\rho$ is the initial-state distribution, $p$ is the transition kernel, and $\gamma \in [0, 1)$ is the discount factor. Offline RL assumes access only to a fixed dataset $\mathcal{D} = \{(s, a, r, s')\}$ collected by an unknown behavior policy, and prohibits additional environment interaction during training. The goal is to learn a parameterized policy $\pi_\theta(a|s)$ that maximizes the expected discounted return under the MDP dynamics:

$$\mathcal{J}(\pi_\theta) = \mathbb{E}_{(s_t, a_t) \sim p^{\pi_\theta}} \left[ \sum_{t=0}^{\infty} \gamma^t r(s_t, a_t) \right], \quad (1)$$

where $p^{\pi_\theta}$ denotes the trajectory distribution induced by $\rho$, $p$, and $\pi_\theta$. To mitigate distribution shift between $\pi_\theta$ and the data-collecting behavior, many offline methods optimize a regularized variant of $\mathcal{J}(\pi_\theta)$ that constrains the learned policy toward actions supported by $\mathcal{D}$. A common perspective is to view this step as constrained policy extraction from an estimated value function, which can be instantiated in several ways (Park et al., 2024):

(i) Behavior-constrained policy gradient:

$$\pi = \arg \max_\pi \mathbb{E}_{\substack{(s,a,r,s') \sim \mathcal{D} \\ a^\pi \sim \pi(\cdot|s)}} \left[ \alpha Q(s, a^\pi) + \log \pi(a|s) \right]. \quad (2)$$

(ii) Weighted behavioral cloning:

$$\pi = \arg \max_\pi \mathbb{E}_{(s,a,r,s') \sim \mathcal{D}} \left[ e^{\alpha(Q(s,a) - V(s))} \log \pi(a|s) \right]. \quad (3)$$

(iii) Sampling-based action selection:

$$\pi(s) = \arg \max_{a_i \sim \pi_{\text{BC}}(\cdot|s), i \in \{1,2,...,N\}} Q(s, a_i),$$
$$\pi_{\text{BC}} = \arg \max_\pi \mathbb{E}_{(s,a,r,s') \sim \mathcal{D}} \left[ \log \pi(a|s) \right]. \quad (4)$$

Here $Q$ denotes the state-action value function and $V$ denotes the state value function. In practice, they are typically estimated from the offline dataset using Monte Carlo return estimation or temporal-difference (TD) learning. The hyperparameter $\alpha$ and the number of candidates $N$ control how strongly the extracted policy is constrained to remain within the dataset support.

From the objectives above, we see that in (ii) and (iii), the value functions $Q$ and $V$ only serve to reweight behavioral samples or select the optimal action from a set of behavior cloning (BC) produced candidates. As a result, the optimization largely preserves the underlying structure of BC (note that the BC form here is written using the standard log-likelihood for clarity; our flow matching BC objective takes a different form, which we detail in section 2.2). In contrast, (i) augments behavior cloning with a direct value maximization term $Q(s, a^\pi)$. When instantiated with multi-step flow matching generation, this term raises the BPTT stability issue. Nevertheless, motivated by the empirical evidence that behavior-constrained policy gradient style extraction is often the most effective among these paradigms (Park et al., 2025b), we aim to follow (i) while addressing its BPTT difficulty in flow matching architectures.

## 2.2. Flow Matching and Flow Policies in RL

Flow Matching belongs to a class of generative models that transform a simple prior distribution $p_0$ into a complex data distribution $p_1$ by constructing deterministic trajectories for samples. Rather than directly modeling the density, flow matching learns a time-dependent velocity field $v_\theta(t, x)$ that induces a flow map $\phi_t : \mathbb{R}^d \to \mathbb{R}^d$, representing the evolution of a sample from its initial state $x_0 \sim p_0$ to its state $x_t$ at time $t$. The velocity field prescribes the instantaneous direction of movement for each particle via the Ordinary Differential Equation (ODE):

$$\frac{\mathrm{d}}{\mathrm{d}t} x_t = v_\theta(x_t, t), \quad \text{subject to } x_t = \phi_t(x_0). \quad (5)$$

By integrating these local velocity vectors from $t = 0$ to $t = 1$, the model pushes the initial distribution $p_0$ through a sequence of intermediate states, eventually reshaping it into the target data distribution $p_1$.

Formally, let $x_0 \sim p_0$ be a sample from a simple prior (e.g., a standard Gaussian) and $x_1 \sim p_1$ be a sample from the target data distribution. We use the standard linear interpolation path commonly adopted in flow matching and rectified flow (Liu et al., 2023):

$$x_t = (1 - t)x_0 + tx_1, \qquad t \in [0, 1]. \quad (6)$$

The conditional velocity that drives this motion is $\dot{x}_t = (x_1 - x_0)/(1 - 0)$. Following (Lipman et al., 2023), the standard approach to training $v_\theta$ is to minimize the flow matching objective:

$$\min_\theta \mathbb{E}_{\substack{x_0 \sim p_0, \\ x_1 \sim p_1, \\ t \sim \mathrm{Unif}(0,1)}} \left[ \|v_\theta(x_t, t) - (x_1 - x_0)\|_2^2 \right]. \quad (7)$$

To parameterize a generative policy $\pi_\theta(a|s)$, we extend the flow matching framework from the general coordinate space to the action space by identifying the latent particle $x_t$ in

Equation 5 with a time-dependent action $a_t \in \mathcal{A}$. Unlike the standard formulation, the velocity field in RL must be conditioned on the state $s$ to account for the environmental context. Consequently, we define a state-conditioned velocity field $v_\theta(s, a_t, t)$, which learns to transport a noise particle $a_0 \sim \mathcal{N}(0, I_d)$ to a terminal action $a_1$ at $t = 1$. Here, $a_1$ represents the actual action $a$ to be executed, effectively defining the policy $\pi_\theta(a|s)$. The action generation process is governed by the following state-dependent ODE:

$$\frac{\mathrm{d}}{\mathrm{d}t} a_t = v_\theta(s, a_t, t). \quad (8)$$

In practice, sampling an action entails integrating the state-conditioned velocity field along the temporal path from $t = 0$ to $t = 1$. Using a $K$-step Euler discretization with a uniform step size $\Delta t = 1/K$, the discrete sampling trajectory is iteratively computed as:

$$a_{k+1} = a_k + \Delta t \cdot v_\theta(s, a_k, t_k), \quad (9)$$

where $t_k = k/K$ and the final action is taken as $a = a_K$. Our goal is to optimize $\pi_\theta$ according to the composite objective in Equation 2. In our framework, the BC component is instantiated by the flow matching objective. However, a discrepancy arises: while the BC signal provides supervision across all $t$, the RL objective $Q(s, a)$ is a terminal signal evaluated only at the end of the sampling trajectory. Naively differentiating this terminal value through the recursive updates in Equation 9 via BPTT requires propagating gradients through long sequences of dependencies (Park et al., 2025b; Zhang et al., 2025c). As discussed in the section 3.1, this leads to optimization instability, necessitating a more robust way to inject Q-gradients into the velocity field.

## 3. Direct Flow Q-Learning

To bridge the gap between velocity-space modeling and action-space RL optimization, we introduce Direct Flow Q-Learning (DFQL). Unlike existing methods that rely on distillation or complex architectural modifications, DFQL optimizes the flow matching policy by injecting value gradients directly into the velocity field. We first formalize the optimization challenges that arise when applying standard RL objectives to iterative flow matching policies.

### 3.1. Instability of Terminal Q-Signal BPTT in Flow Policies

By temporarily setting aside the BC constraint and focusing exclusively on the RL objective of Q-value maximization, while incorporating the iterative generation process of flow matching, we can derive the RL-specific optimization objective as follows:

$$\max_\theta \ \mathcal{J}(\theta) = \mathbb{E}_{\substack{(s,a,r,s') \sim \mathcal{D} \\ a_0 \sim \mathcal{N}(0, I_d)}} \left[ Q(s, a_K(s, a_0; \theta)) \right]. \quad (10)$$

To optimize this objective, the gradient $\nabla_\theta \mathcal{J}$ must be propagated back from the terminal action $a_K$ through the entire sampling chain. Applying the chain rule, we obtain:

$$\nabla_\theta \mathcal{J}(\theta) = \mathbb{E}_{\substack{(s,a,r,s') \sim \mathcal{D} \\ a_0 \sim \mathcal{N}(0,I_d)}} \left[ \nabla_{a_K} Q(s, a_K) \cdot \frac{\partial a_K}{\partial \theta} \right] \quad (11)$$

The term $\frac{\partial a_K}{\partial \theta}$ encapsulates the sensitivity of the final action to the policy parameters, which can be expanded recursively:

$$\frac{\partial a_K}{\partial \theta} = \sum_{k=0}^{K-1} \left( \prod_{j=k+1}^{K-1} (I + \Delta t \cdot J_j) \right) \Delta t \frac{\partial v_\theta(s, a_k, t_k)}{\partial \theta},$$
$$J_j \triangleq \frac{\partial v_\theta(s, a_j, t_j)}{\partial a_j}. \quad (12)$$

This recursive formulation reveals the gradient bottleneck: the gradient at early steps is scaled by a product of Jacobians $J_j$. In deep flow architectures, the term $\prod(I + \Delta t \cdot J_j)$ may grow exponentially with $K$, leading to numerical instability.

### 3.2. Our Solution

The core philosophy of DFQL is to transform the terminal RL signal into a local, step-wise supervision for the velocity field. This design follows the flow-matching principle of learning local velocity fields along a probability path, while adapting it to the RL setting where policy improvement is specified by terminal value guidance rather than explicit target actions at intermediate steps. Instead of backpropagating through the entire sampling chain, we use the terminal Q-gradient to guide the velocity at each integration step toward regions of higher value.

For a given $K$-step Euler-discretized integration path $\{a_0, a_1, \ldots, a_K\}$, let $g = \nabla_{a_K} Q(s, a_K)$ denote the terminal Q-gradient. We aim to adjust the velocity $v_\theta(s, a_k, t_k)$ such that it nudges the particle in a direction that maximizes the action value. We define the "value-improved" (RL) velocity target as:

$$\tilde{v}_k = v_\theta(s, a_k, t_k) + \Delta t \cdot g. \quad (13)$$

To minimize the discrepancy between the current velocity field and this improved target, we consider the following mean squared error loss at each step $k$:

$$\mathcal{L}_k(\theta) = \frac{1}{2} \| v_\theta(s, a_k, t_k) - \text{sg}[\tilde{v}_k] \|_2^2, \quad (14)$$

where $\text{sg}[\cdot]$ denotes the stop-gradient operator.

Although Equation 14 is formulated as a regression task, we show that in terms of the gradient with respect to $\theta$, it is equivalent to a linear projection of the velocity field onto the Q-gradient. Specifically:

$$\nabla_\theta \mathcal{L}_k(\theta) = \nabla_\theta \left( -\Delta t \cdot g^\top v_\theta(s, a_k, t_k) \right). \quad (15)$$

This equivalence allows us to bypass the quadratic term and directly optimize the inner product between the velocity field and the guidance signal. We provide a rigorous derivation of this gradient equivalence in Appendix A.1.

Since the terminal gradient $g$ is evaluated at $t = 1$, its reliability in guiding the velocity field varies across the trajectory. Intuitively, velocities at later timesteps have a more direct impact on the final action $a_K$. We thus introduce a temporal weight $w_k = 1/(K-k)$ to reflect this increasing importance. Summing over the $K$-step sampling chain, the final RL objective for DFQL is:

$$\mathcal{L}_{\text{RL}}(\theta) = -\mathbb{E}_{\substack{(s,a,r,s') \sim \mathcal{D} \\ a_0 \sim \mathcal{N}(0,I_d)}} \left[ \sum_{k=0}^{K-1} \left( w_k \Delta t \cdot g^\top v_\theta(s, a_k, t_k) \right) \right]. \quad (16)$$

By distributing the terminal reward as step-wise velocity guidance, DFQL effectively internalizes value-awareness into the flow-matching policy without the numerical instability of BPTT.

**Explanation: Why This Works?**

To investigate the efficacy of DFQL, we examine the gradient of the terminal value $Q(s, a_K)$ with respect to the velocity parameters $\theta$ at an intermediate sampling step $k \in \{0, \ldots, K-1\}$. For a given trajectory, applying BPTT through the discretized path yields:

$$\nabla_\theta Q_{k,\text{BPTT}} = g^\top \underbrace{\left[ \prod_{j=k+1}^{K-1} (I + \Delta t \cdot J_j) \right]}_{\text{Backpropagated Signal}} \cdot \underbrace{\Delta t \frac{\partial v_\theta(s, a_k, t_k)}{\partial \theta}}_{\text{Local Sensitivity}}. \quad (17)$$

The term $I + \Delta t \cdot J_j$ implies that the gradient is not merely a product of deep Jacobians but travels through an explicit identity pathway at each step. In flow matching policies, the discretization step $\Delta t$ acts as a natural regularizer. For a Lipschitz-continuous $v_\theta$, the per-step Jacobian $I + \Delta t \cdot J_j$ remains $O(\Delta t)$ close to $I$. This ensures the terminal guidance $g$ remains a dominant signal throughout the Euler-discretized integration path even without full Jacobian propagation. DFQL effectively performs state-sensitivity pruning by setting $\Delta t \cdot J_j \approx 0$ during gradient computation. This isolates the stable identity pathway and discards the interaction terms $\Delta t \cdot J_j$, which are the primary sources of exploding gradients in long-horizon offline RL. The resulting gradient contribution from step $k$ in the DFQL objective is:

$$\nabla_\theta Q_{k,\text{DFQL}} = w_k \cdot g^\top \cdot \Delta t \frac{\partial v_\theta(s, a_k, t_k)}{\partial \theta}. \quad (18)$$

While this introduces a bias by omitting inter-step dependencies, we compensate for the accumulated error by using the increasing temporal weight $w_k = 1/(K-k)$, prioritizing steps closer to the terminal evaluation where the approximation is most accurate. In Appendix A.2, we demonstrate

the asymptotic consistency of this objective, proving that as $\Delta t \to 0$, the terminal-action update direction induced by DFQL RL component aligns with the terminal guidance direction, serving as a principled first-order approximation for iterative policies.

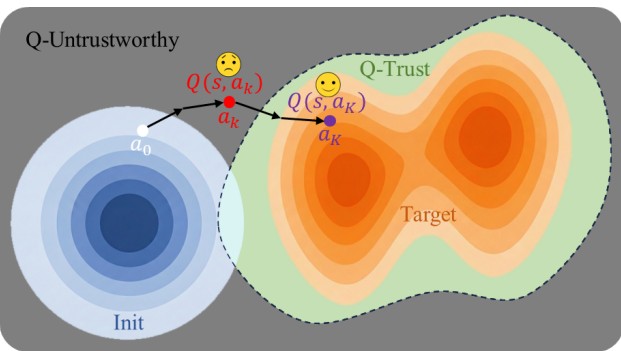

*Figure 2.* As intermediate actions $a_k$ may stray far from the target distribution, the resulting $Q(s, a_k)$ becomes untrustworthy, compromising the accuracy of the value evaluation and subsequent updates.

Beyond its stability as a BPTT approximation, our objective can be viewed as moving inference-time guidance (Jang et al., 2025) into the training objective. Standard guidance techniques typically require computing $\nabla_{a_k} Q(s, a_k)$ at every sampling step to shift the trajectory, which incurs computational overhead during inference. Furthermore, in offline RL, the critic Q is trained on the data distribution of terminal actions. Evaluating its gradients on intermediate, "half-finished" particles $a_t$ may lead to unreliable updates due to OOD states. In contrast, DFQL only utilizes the gradient evaluated at the terminal action $a_K$, where the critic is most accurate.

### 3.3. Practical Implementation and Global Objective

To instantiate the DFQL framework, we now describe the learning of the critic, the refined behavior cloning objective, and the unified optimization pipeline.

**Critic learning.** DFQL adopts a standard TD learning objective to train the critic. Given an offline dataset $\mathcal{D}$, we learn a parametric state-action value function $Q_\phi(s, a)$. The critic is optimized by minimizing the squared TD error:

$$
\begin{aligned}
\mathcal{L}_Q(\phi) &= \mathbb{E}_{(s,a,r,s') \sim \mathcal{D}} \left[ (Q_\phi(s, a) - y)^2 \right], \\
y &= r + \gamma \mathbb{E}_{a' \sim \pi_\theta(\cdot | s')} \left[ Q_{\bar{\phi}}(s', a') \right],
\end{aligned}
\tag{19}
$$

where $\bar{\phi}$ denotes the parameters of the target network. The expectation over next actions $a'$ is computed by passing noise through the multi-step flow policy. Unless otherwise specified, we follow standard offline RL practices, including the use of target networks and double critics.

**BC constraint.** While standard flow matching uniformly samples $t \in [0, 1]$, we propose a dense BC objective to

---

**Algorithm 1:** Direct Flow Q-Learning (DFQL)

**Function** $\mu_\theta (s, z)$
   $a_0 \leftarrow z$
   **for** $k = 0, 1, \ldots, K - 1$ **do**
      $a_{k+1} \leftarrow a_k + \frac{1}{K} v_\theta(s, a_k, t_k)$
   **return** $(a_0, a_1, \ldots, a_K)$

**while** *not converged* **do**
   Sample batch $(s, a, r, s') \sim \mathcal{D}$; $z_c, z_a \sim \mathcal{N}(0, I_a)$

   $(a_0, a_1, \ldots, a_K) \leftarrow \mu_\theta(s', z_c)$      ▷ Critic learning
   $a' \leftarrow a_K$
   Train critic $Q_\phi$ by minimizing Equation 19

   $(a_0, a_1, \ldots, a_K) \leftarrow \mu_\theta(s, z_a)$      ▷ Actor learning
   $g \leftarrow \nabla_{a_K} Q_\phi(s, a_K)$
   Compute BC loss $\mathcal{L}_{\text{BC}}(\theta)$ using Equation 20
   Compute RL loss $\mathcal{L}_{\text{RL}}(\theta)$ using Equation 16
   Compute combine loss $\mathcal{L}_\pi(\theta) = \mathcal{L}_{\text{BC}}(\theta) + \alpha \mathcal{L}_{\text{RL}}(\theta)$
   Train actor $\pi_\theta$ by minimizing Equation 21

**return** policy $\pi_\theta$

---

ensure structural symmetry with our RL objective. Since $\mathcal{L}_{\text{RL}}$ provides supervision at each discrete time step $k$, a sparse BC signal may lead to temporal imbalance in the supervision provided to the velocity field. We thus train $v_\theta$ on each discrete point along the straight-line bridge:

$$
\mathcal{L}_{\text{BC}}(\theta) = \mathbb{E}_{\substack{(s,a,r,s') \sim \mathcal{D} \\ a_0^{\text{BC}} \sim \mathcal{N}(0, I_d)}} \left[ \sum_{k=0}^{K-1} \| v_\theta(s, a_k^{\text{BC}}, t_k) - (a - a_0^{\text{BC}}) \|_2^2 \right],
\tag{20}
$$

where $a_k^{\text{BC}} = (1 - t_k) a_0^{\text{BC}} + t_k a$ represents the interpolated particle at step $k$. This dense supervision ensures that the imitation and reinforcement signals operate at the same "density" in velocity space, enhancing training stability when the two objectives are combined.

**Unified policy optimization.** The final policy objective balances the refined BC loss with the proposed DFQL RL loss. By combining Equation 16 and Equation 20, the global optimization problem is formulated as:

$$
\mathcal{L}_\pi(\theta) = \mathcal{L}_{\text{BC}}(\theta) + \alpha \mathcal{L}_{\text{RL}}(\theta),
\tag{21}
$$

where $\alpha$ is a hyperparameter that controls the trade-off between maximizing the Q-value and adhering to the data distribution. The complete training procedure is detailed in Algorithm 1, and more detailed implementation specifics are provided in Appendix C.

# 4. Related Works

## 4.1. Offline RL and Policy Extraction

Offline RL aims to derive effective policies from static datasets while mitigating performance collapse caused by OOD actions (Levine et al., 2020). This is typically achieved through conservative value learning (Kumar et al., 2020; Yu et al., 2021; An et al., 2021) or constrained policy extraction (Wang et al., 2025). As formalized in Section 2, existing policy extraction paradigms can be broadly categorized into three lines. **Behavior-constrained policy gradient** methods directly maximize the Q-function alongside a BC regularizer (Fujimoto & Gu, 2021; Tarasov et al., 2023a; Liu et al., 2024); while these methods (e.g., TD3+BC) are highly effective, they encounter significant optimization hurdles when applied to multi-step generative policies due to the need for differentiation through the generated trajectory (Fang et al., 2025). Alternatively, **weighted behavioral cloning** approaches, such as AWR or IQL, reweight dataset actions based on value or advantage estimates (Nair et al., 2020; Qing et al., 2024; Peters & Schaal, 2007). This paradigm is naturally stable and compatible with generative models but is often overly conservative as policy improvement is strictly bounded by the dataset support (Kostrikov et al., 2022). Lastly, **sampling-based selection** methods utilize the critic to rank and select candidates from multiple action proposals (Kang et al., 2023; Mao et al., 2024; Chen et al., 2023). While these are compatible with expressive policy parameterizations, they incur significant computational costs during training and inference due to the extensive sampling required (Fujimoto et al., 2019; Chen et al., 2024b). Our framework, DFQL, builds on the behavior-constrained paradigm but circumvents its optimization instability by directly injecting value gradients into the velocity field.

## 4.2. Flow-Matching and other Generative Policies

Generative policies are typically more expressive than simple unimodal parameterizations, enabling the modeling of multimodal action distributions. In offline settings, this can facilitate better alignment with the dataset support. Prior work has explored generative policies along multiple technical directions, including parameterizations based on latent variables (Zhou et al., 2020), autoregressive distributions (Korenkevych et al., 2019), mixture models (Zhao et al., 2022), and trajectory modeling (Chen et al., 2021). Recently, iterative generative policies have attracted increasing attention, with diffusion models (Wang et al., 2023; Chen et al., 2024b; Fang et al., 2025; Mao et al., 2024) and flow matching emerging as prominent routes for bringing highly expressive generative parameterizations to offline RL.

Motivated by the simplicity of flow matching architectures, their often higher inference efficiency relative to diffusion-based policies (Park et al., 2025b; Kang et al., 2023), and

the fact that this direction remains comparatively under-explored, offline RL with flow matching policy networks has received increasing attention. Our work also falls into this category. Among existing work, a representative line, exemplified by FQL, distills a multi-step generative policy into a one-step policy, providing a stable interface for value-based policy improvement (Park et al., 2025b; Tiofack et al., 2025). Another line incorporates critic preferences as energy weighting or guidance, injecting them into the flow matching objective or the sampling procedure to drive policy improvement (Feng et al., 2025; Zhang et al., 2025a; Alles et al., 2025). Beyond these, recent studies explore path-level regularization to improve the stability of learning the vector field (Chen et al., 2025b), employ specialized network architectures such as gating mechanisms to stabilize the BPTT process (Zhang et al., 2025c), or explicitly formulate the generation process as a sequential decision problem (Zhang et al., 2025b). DFQL is most closely related to FQL, but does not rely on policy distillation. Instead, it stabilizes training via a surrogate objective and applies RL-based improvement directly to the flow matching network. Beyond purely offline RL, many studies also investigate flow matching policies in the offline-to-online setting (Shin et al., 2026; Li et al., 2025). Our experiments show that DFQL can be applied to offline-to-online RL without any modification and still meets the requirements of this setting.

# 5. Experiments

In this section, we evaluate our method and compare it against prior offline RL and offline-to-online RL approaches across challenging tasks. We also run ablations to examine key design choices and strategy components.

## 5.1. Experimental Setup

**Benchmarks.** We evaluate our method on two standard offline RL benchmarks, OGBench (Park et al., 2025a) and D4RL (Fu et al., 2020). Our main evaluation is on OG-Bench, which offers a broad collection of locomotion and manipulation tasks and supports state-based settings under a unified interface. Following the established protocol and keeping the setup consistent with FQL(Park et al., 2025b), we use the reward-based single-task variants of OGBench and report results on 50 state-based tasks. We allocate a larger portion of experiments to OGBench to assess performance across a wider range of task conditions and behavioral coverage, enabling more informative comparisons under demanding settings. In parallel, we retain evaluation on D4RL to improve comparability with prior work, focusing on the commonly reported AntMaze navigation suite and the Adroit dexterous manipulation suite.

**Methods compared.** In the offline RL experiments, we compare against ten methods spanning three categories:

*Table 1.* **Offline RL results.** Across the vast majority of environments, DFQL achieves optimal or near-optimal performance, demonstrating strong and consistent results. All numbers are reported as means over 8 seeds (4 seeds for pixel-based tasks), and the complete experimental results are provided in Table 5.

| Task category | Gaussian policies | | | Diffusion policies | | | Flow policies | | | | |
|---|---|---|---|---|---|---|---|---|---|---|---|
| | BC | IQL | ReBRAC | IDQL | SRPO | CAC | FAWAC | FBRAC | IFQL | FQL | DFQL |
| OGbench antmaze-large-singletask (**5** tasks) | $11\pm1$ | $53\pm3$ | $81\pm5$ | $21\pm5$ | $11\pm4$ | $33\pm4$ | $6\pm1$ | $60\pm6$ | $28\pm5$ | $79\pm3$ | $\mathbf{88}\pm2$ |
| OGbench antmaze-giant-singletask (**5** tasks) | $0\pm0$ | $4\pm1$ | $26\pm8$ | $0\pm0$ | $0\pm0$ | $0\pm0$ | $0\pm0$ | $4\pm4$ | $3\pm2$ | $9\pm6$ | $\mathbf{39}\pm4$ |
| OGbench humanoidmaze-medium-singletask (**5** tasks) | $2\pm1$ | $33\pm2$ | $22\pm8$ | $1\pm0$ | $1\pm1$ | $53\pm8$ | $19\pm1$ | $38\pm5$ | $60\pm14$ | $58\pm5$ | $\mathbf{66}\pm9$ |
| OGbench humanoidmaze-large-singletask (**5** tasks) | $1\pm0$ | $2\pm1$ | $2\pm1$ | $1\pm0$ | $0\pm0$ | $0\pm0$ | $0\pm0$ | $2\pm0$ | $\mathbf{11}\pm2$ | $4\pm2$ | $8\pm3$ |
| OGbench antsoccer-arena-singletask (**5** tasks) | $1\pm0$ | $8\pm2$ | $0\pm0$ | $12\pm4$ | $1\pm0$ | $2\pm4$ | $12\pm0$ | $16\pm1$ | $33\pm6$ | $\mathbf{60}\pm2$ | $59\pm4$ |
| OGbench cube-single-singletask (**5** tasks) | $5\pm1$ | $83\pm3$ | $91\pm2$ | $95\pm2$ | $80\pm5$ | $85\pm9$ | $81\pm4$ | $79\pm7$ | $79\pm2$ | $\mathbf{96}\pm1$ | $\mathbf{96}\pm1$ |
| OGbench cube-double-singletask (**5** tasks) | $2\pm1$ | $7\pm1$ | $12\pm1$ | $15\pm6$ | $2\pm1$ | $6\pm2$ | $5\pm2$ | $15\pm3$ | $14\pm3$ | $29\pm2$ | $\mathbf{38}\pm7$ |
| OGbench scene-singletask (**5** tasks) | $5\pm1$ | $28\pm1$ | $41\pm3$ | $46\pm3$ | $20\pm1$ | $40\pm7$ | $30\pm3$ | $45\pm5$ | $30\pm3$ | $56\pm2$ | $\mathbf{61}\pm3$ |
| OGbench puzzle-3x3-singletask (**5** tasks) | $2\pm0$ | $9\pm1$ | $21\pm1$ | $10\pm2$ | $18\pm1$ | $19\pm0$ | $6\pm2$ | $14\pm4$ | $19\pm1$ | $30\pm1$ | $\mathbf{65}\pm7$ |
| OGbench puzzle-4x4-singletask (**5** tasks) | $0\pm0$ | $7\pm1$ | $14\pm1$ | $\mathbf{29}\pm3$ | $10\pm3$ | $15\pm3$ | $1\pm0$ | $13\pm1$ | $25\pm5$ | $17\pm2$ | $25\pm3$ |
| D4RL antmaze (**6** tasks) | $17$ | $57$ | $78$ | $79$ | $74$ | $30$ | $44\pm3$ | $64\pm7$ | $65\pm7$ | $\mathbf{84}\pm3$ | $82\pm3$ |
| D4RL adroit (**12** tasks) | $48$ | $53$ | $\mathbf{59}$ | $52\pm1$ | $51\pm1$ | $43\pm2$ | $48\pm1$ | $50\pm2$ | $52\pm1$ | $52\pm1$ | $52\pm1$ |
| Visual manipulation (**5** tasks) | - | $42\pm4$ | $60\pm2$ | - | - | - | - | $22\pm2$ | $50\pm5$ | $65\pm2$ | $\mathbf{72}\pm5$ |

*Table 2.* **Offline-to-online RL results.** Consistent with established benchmarks for offline-to-online RL (Prudencio et al., 2024), we employ 1M steps for both the initial offline phase and the subsequent online fine-tuning stage. Full results are presented in Table 4.

| BC | IQL | ReBRAC | Cal-QL | RLPD | IFQL | FQL | DFQL |
|---|---|---|---|---|---|---|---|
| OGbench (**5** tasks)[1] | $8\pm3 \rightarrow 5\pm2$ | $17\pm5 \rightarrow 29\pm9$ | $0\pm0 \rightarrow 10\pm11$ | $0\pm0 \rightarrow 42\pm2$ | $23\pm8 \rightarrow 48\pm11$ | $40\pm4 \rightarrow 68\pm11$ | $48\pm7 \rightarrow \mathbf{89}\pm9$ |
| D4RL antmaze (**6** tasks) | $56 \rightarrow 78$ | $80 \rightarrow 79$ | $53 \rightarrow \mathbf{97}$ | $0\pm0 \rightarrow 96\pm1$ | $65\pm6 \rightarrow 90\pm2$ | $75\pm8 \rightarrow \mathbf{95}\pm5$ | $83\pm2 \rightarrow \mathbf{96}\pm1$ |
| D4RL adroit (**12** tasks) | $22 \rightarrow 45$ | $21 \rightarrow 93$ | $-1 \rightarrow -1$ | $1\pm1 \rightarrow 88\pm8$ | $21\pm2 \rightarrow 56\pm6$ | $13\pm4 \rightarrow \mathbf{110}\pm5$ | $17\pm2 \rightarrow \mathbf{106}\pm4$ |

[1]As offline-to-online RL is not the primary focus of our study, we evaluate only five representative environments from the OGBench suite to ensure a consistent experimental setup with FQL (Park et al., 2025b).

Gaussian, diffusion, and flow-based approaches. The Gaussian category includes the basic BC baseline, the classic IQL (Kostrikov et al., 2022), and the strong performer ReBRAC (Tarasov et al., 2023a). The diffusion category covers IDQL (Hansen-Estruch et al., 2023) based on rejection sampling, as well as SRPO (Chen et al., 2024a) and CAC (Ding & Jin, 2024) based on policy distillation. The flow category includes representative flow-policy methods that instantiate different policy extraction strategies on top of a flow matching policy, namely FAWAC as a flow variant of AWAC (Nair et al., 2020), IFQL as a flow variant of rejection-sampling methods, and FBRAC as a flow variant of BRAC (Wu et al., 2019), together with the high-performing policy-distilled flow matching method FQL. These flow-based comparisons are designed to highlight how different policy extraction strategies behave when combined with flow matching in a straightforward manner. In particular, FBRAC is the closest baseline to our DFQL, but it relies on the classical gradient backpropagation mechanism to propagate terminal Q information through all discrete generation steps. Contrasting it with our direct guidance injection provides an ablation-style evaluation of the proposed information injection design.

Beyond comparisons with purely offline methods, we also evaluate our approach in the offline-to-online RL setting. In this study, we include three strong baselines that support offline-to-online fine-tuning (IQL, ReBRAC, and FQL) and additionally compare against two methods specifically designed for offline-to-online RL, Cal-QL (Nakamoto et al., 2023) and RLPD (Ball et al., 2023).

**Evaluation.** We evaluate all methods after a fixed number of gradient update steps. To mitigate potential reporting bias, we report the final performance rather than the best score observed during training (Tarasov et al., 2023b). Results are presented as the mean and variability across multiple random seeds, values following "±" in tables denote standard deviation, while shaded areas in plots represent the 95% bootstrap confidence interval. To ensure a fair comparison, we adopt the baseline results directly from FQL (Park et al., 2025b), where hyperparameters were extensively optimized. For DFQL, we follow the same MLP backbone configuration and discount factor as in that study to ensure comparability, while any minor implementation deviations are described in Appendix C. Regarding evaluation metrics, OGBench tasks are measured by the episode success rate, while on D4RL we follow the standard protocols, using binary success rates for AntMaze and normalized returns for Adroit. Full training and evaluation details are provided in Appendix C and F.

## 5.2. Main Results and Analysis

**Performance comparison.** According to the comparative results in Table 1, DFQL achieves superior performance over both existing strong Gaussian-policy methods and diffusion-policy methods, demonstrating the effectiveness of the flow matching policy itself. Moreover, DFQL's advantage over FBRAC indicates that the approximations adopted in our approach effectively stabilize the training process. Its improvement over FQL further suggests that retaining the multi-step generation procedure indeed leads to performance gains. Notably, the substantial improvements of DFQL on challenging tasks (e.g., "antmaze-giant-singletask" and "puzzle-3x3-singletask-task") reflect enhanced representational capacity for complex tasks. In summary, the experimental results demonstrate that DFQL is a viable and competitive method for offline RL.

The results for the offline-to-online transition are reported in Table 2. Notably, our method achieves optimal or near-optimal performance in this phase even without incorporating specialized offline-to-online mechanisms.

**Effectiveness of DFQL in mitigating gradient explosion.** To isolate DFQL's effect on gradient explosion, we replace the terminal-gradient direct guidance with a BPTT-style step-by-step backpropagation baseline while keeping all other settings unchanged. We then evaluate training stability by tracking the maximum gradient magnitude on three environments. As shown in Figure 3, DFQL substantially mitigates the gradient explosion under BPTT, leading to a more stable optimization process. Additional ablations on other key components are provided in Appendix C.

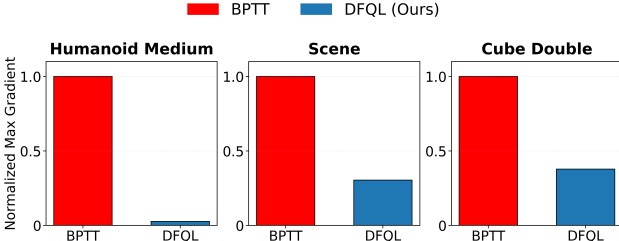

*Figure 3.* **Comparison of maximum gradient magnitudes.** To represent the worst-case scenario, we collect the maximum absolute gradients across 8 seeds and normalize the values within each task by scaling the largest magnitude to 1.0.

**DFQL as a modular algorithmic component.** Beyond its standalone performance, we investigate whether the gradient stability provided by DFQL can be combined with other orthogonal RL enhancements. To evaluate this, we integrate our method with Q-chunking (QC), a technique specifically tailored for long-horizon and sparse-reward tasks by operating within a temporally extended action space (Li et al., 2025). Specifically, QC trains the policy to predict a sequence of future actions while the critic estimates the

value of the entire sequence, promoting temporally coherent exploration and enabling unbiased $n$-step value backups. Crucially, DFQL and QC target different stages of the RL pipeline. Our method stabilizes the policy extraction process from flow-matching-based models, whereas QC focuses on optimizing action representation and credit assignment. This structural difference allows for a seamless integration into the QC-DFQL variant. As illustrated in Figure 4, QC-DFQL achieves a consistent performance boost over the high-performing QC-FQL (a variant combining QC with FQL) baseline across representative sparse-reward manipulation tasks. These results suggest that DFQL is a lightweight modification that can synergize with other RL architectures to further enhance performance in complex environments. We provide a further introduction of the QC-DFQL implementation in Appendix D.

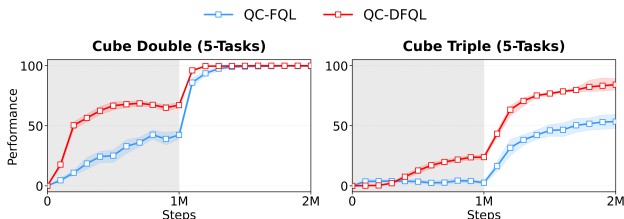

*Figure 4.* **Synergy between DFQL and QC.** To evaluate DFQL's modularity, we integrate it with QC across "cube-double/triple" tasks. Results show a two-stage training trajectory: offline RL (0-1M steps) and online fine-tuning (1-2M steps), with the transition marked by the shaded boundary. QC-DFQL outperforms the QC-FQL baseline, highlighting DFQL as a compatible component for flow-matching-based RL architectures. Full results in Fig. 7.

## 6. Conclusion

In this paper, we presented Direct Flow Q-Learning, an approach to integrating flow matching with offline RL. By addressing the fundamental challenges of BPTT-based optimization, we provide a compelling alternative to the common distillation-centric framework. Our key contribution is the development of a surrogate objective that enables the direct propagation of RL signals into the flow matching vector field. This mechanism facilitates stable and efficient policy learning while maintaining the multi-step sampling structure, thereby avoiding the potential trade-offs associated with one-step approximations. Extensive empirical evaluations across 73 tasks confirm that DFQL not only achieves superior performance in static offline datasets but also exhibits robust adaptability during offline-to-online transition. While our work demonstrates significant advancements, a discussion regarding its current limitations is provided in Appendix B. We believe that the paradigm of direct gradient injection offered by DFQL provides a scalable and effective blueprint for future research in flow-matching-based RL.

## Acknowledgments

This work was supported by the National Natural Science Foundation of China under Grant No. 62303244, and the Open Fund Project of the State Key Laboratory of Intelligent Green Vehicle and Mobility under Grant No. KFY260406.

## Impact Statement

This paper presents work whose goal is to advance the field of Machine Learning. There are many potential societal consequences of our work, none which we feel must be specifically highlighted here.

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

## A. Derivations

### A.1. Gradient equivalence for RL loss

This section provides a formal derivation showing that the gradient of the quadratic loss in Equation 14 is equivalent to the gradient of the inner product objective in Equation 15. Starting from the loss function $\mathcal{L}_k(\theta)$ defined in Equation 14:

$$\mathcal{L}_k(\theta) = \frac{1}{2}\|v_\theta(s, a_k, t_k) - \text{sg}[\tilde{v}_k]\|^2 \tag{22}$$

where $\text{sg}[\cdot]$ is the stop-gradient operator. Let $v_\theta$ denote $v_\theta(s, a_k, t_k)$ for brevity. The gradient of this scalar loss with respect to the parameters $\theta$ is computed as:

$$\nabla_\theta \mathcal{L}_k(\theta) = (v_\theta - \text{sg}[\tilde{v}_k])^\top \nabla_\theta v_\theta. \tag{23}$$

By the definition of the stop-gradient operator, $\text{sg}[\tilde{v}_k]$ is treated as a constant vector during differentiation. According to Equation 13, we have $\tilde{v}_k = v_\theta + \Delta t \cdot g$. When evaluating the gradient at the current parameter state $\theta$, the value inside the stop-gradient operator matches the current output of the network. We can denote this fixed value as $v_{\text{fixed}} = v_\theta|_\theta$. Thus:

$$\nabla_\theta \mathcal{L}_k(\theta) = (v_\theta - (v_{\text{fixed}} + \Delta t \cdot g))^\top \nabla_\theta v_\theta. \tag{24}$$

Expanding the terms, we obtain:

$$\nabla_\theta \mathcal{L}_k(\theta) = (v_\theta - v_{\text{fixed}})^\top \nabla_\theta v_\theta - (\Delta t \cdot g)^\top \nabla_\theta v_\theta. \tag{25}$$

At the point of evaluation, the current velocity $v_\theta$ is numerically equal to the fixed velocity $v_{\text{fixed}}$, which implies that the term $(v_\theta - v_{\text{fixed}})$ becomes zero. Consequently, the first term vanishes:

$$\nabla_\theta \mathcal{L}_k(\theta) = -\Delta t \cdot g^\top \nabla_\theta v_\theta. \tag{26}$$

Since we treat $g$ as a gradient-clipped constant here, both $g$ and $\Delta t$ are independent of $\theta$. Therefore, we can rewrite the expression as the gradient of an inner product:

$$\nabla_\theta \mathcal{L}_k(\theta) = \nabla_\theta \left(-\Delta t \cdot g^\top v_\theta(s, a_k, t_k)\right). \tag{27}$$

This completes the proof that the quadratic loss with a stop-gradient is equivalent to the linear projection-based optimization in terms of gradient descent.

### A.2. Asymptotic consistency of the temporally weighted objective

In this section, we provide a formal proof of the asymptotic consistency for the proposed objective in Equation 16. We demonstrate that, for a Lipschitz-continuous $v_\theta$, by employing the temporal weighting $w_k = 1/(K - k)$, the cumulative change of the terminal state $a_K$ asymptotically aligns with the guidance direction $g$ as the number of discrete steps $K \to \infty$. Our proof first analyzes the contribution of the change at time step $k$ to the terminal state under the loss defined in Equation 16. We then examine how the aggregation of these contributions across all time steps affects the alignment between the resulting terminal-state update direction and the desired direction $g$.

**Variational perturbation in velocity space.** We analyze the optimization through the lens of functional gradient descent in the velocity space. Given the objective $\mathcal{L}_{\text{RL}} = -\sum_{k=0}^{K-1} w_k \Delta t \cdot g^\top v_k$, the steepest descent direction with respect to the velocity field at step $k$ is defined by the functional derivative:

$$-\frac{\delta \mathcal{L}_{\text{RL}}}{\delta v_k} = w_k \Delta t \cdot g. \tag{28}$$

A gradient update with an effective learning rate $\eta > 0$ induces a local velocity perturbation $\delta v_k = \eta w_k \Delta t \cdot g$.

**Propagation through discrete dynamics.** To evaluate the impact of a local perturbation $\delta v_k$ on the terminal state $a_K$, we examine the perturbation propagation through the discrete-time dynamical system. The state at step $k + 1$ is updated as $a_{k+1} = a_k + \Delta t \cdot v_\theta(a_k, t_k)$. The local velocity change $\delta v_k$ induces an immediate displacement:

$$\delta a_{k+1} = \Delta t \cdot \delta v_k = \eta w_k (\Delta t)^2 \cdot g. \tag{29}$$

Consider the propagation of this displacement to the subsequent step $a_{k+2}$. Applying a first-order Taylor expansion to the evolution equation:

$$a_{k+2} + \delta a_{k+2} = (a_{k+1} + \delta a_{k+1}) + \Delta t \cdot v_\theta(s, a_{k+1} + \delta a_{k+1}, t_{k+1}), \tag{30}$$

$$\delta a_{k+2} \approx \delta a_{k+1} + \Delta t \frac{\partial v_\theta}{\partial a}\bigg|_{a_{k+1}} \delta a_{k+1} = (I + \Delta t \cdot J_{k+1})\delta a_{k+1}, \tag{31}$$

where $J_{k+1}$ denotes the Jacobian matrix of the velocity field at step $k+1$. By recursive application, the terminal displacement $\Delta a_K^{(k)}$ originating from step $k$ is given by:

$$\Delta a_K^{(k)} \triangleq \left( \prod_{j=k+1}^{K-1} (I + \Delta t \cdot J_j) \right) \delta a_{k+1} = \Phi_{k \to K} \delta a_{k+1}, \tag{32}$$

where $\Phi_{k \to K}$ is the discrete propagator characterizing the cumulative influence of the system dynamics on the initial perturbation.

**Effective propagator and linearization.** Assuming the velocity field $v_\theta(s, a, t)$ is Lipschitz continuous in $a$, the sequence of Jacobians is bounded, (i.e., $\|J_j\| \le L$). For a sufficiently large $K$ (small $\Delta t = 1/K$), the propagator can be linearized as:

$$\Phi_{k \to K} = I + \Delta t \sum_{j=k+1}^{K-1} J_j + \mathcal{O}(\Delta t^2). \tag{33}$$

We define the effective average Jacobian over the remaining path as $\bar{J}_i = \frac{1}{i} \sum_{j=k+1}^{K-1} J_j$, where $i = K - 1 - k$ is the number of remaining steps. The propagator simplifies to:

$$\Phi_{k \to K} \approx I + \frac{i}{K} \bar{J}_i. \tag{34}$$

This term represents the first-order approximation of the dynamical drift that potentially rotates the intended update direction $g$.

**Asymptotic directional alignment.** The total displacement $\Delta a_{\text{total}}$ is the summation of contributions from all steps $k \in \{0, \dots, K-1\}$. Substituting the specific weighting $w_k = 1/(K-k) = 1/(i+1)$:

$$\Delta a_{\text{total}} = \sum_{i=0}^{K-1} \Delta a_K^{(i)} = \frac{\eta}{K^2} \sum_{i=0}^{K-1} \left[ \frac{1}{i+1} g + \frac{i}{K(i+1)} \bar{J}_i g \right]. \tag{35}$$

We analyze the two components of this summation as $K \to \infty$:

i) The Guidance Term: The first component scales with the harmonic series $\sum_{i=0}^{K-1} \frac{1}{i+1}$, which diverges logarithmically:

$$\left( \sum_{i=0}^{K-1} \frac{1}{i+1} \right) g \to (\ln K)g. \tag{36}$$

ii) The Dynamical Drift Term: The second component represents the accumulated bias from the Jacobians. Under the Lipschitz assumption ($\|\bar{J}_i\| \le L$):

$$\left\| \sum_{i=0}^{K-1} \frac{i}{K(i+1)} \bar{J}_i g \right\| \le \frac{L\|g\|}{K} \sum_{i=0}^{K-1} 1 = L\|g\|. \tag{37}$$

This term is bounded by a constant independent of $K$. Combining these results, the terminal displacement follows the asymptotic form :

$$\Delta a_{\text{total}} \propto (\ln K)g + \mathcal{O}(L)g. \tag{38}$$

As $K \to \infty$, the logarithmic term dominates the constant drift. Consequently, the direction of the total improvement aligns with the guidance signal:

$$\lim_{K \to \infty} \text{cosine similarity}(\Delta a_{\text{total}}, g) = 1. \tag{39}$$

This concludes the proof that, under the Lipschitz continuity assumption on $v_\theta$, the temporal weighting $w_k = 1/(K-k)$ scheme ensures first-order consistency with the terminal objective.

## B. Limitations

While DFQL demonstrates superior performance and stability, we identify several limitations that offer promising directions for future research.

**Inference latency vs. expressivity.** One primary limitation of DFQL is the inherent trade-off between representation capability and computational efficiency. By preserving the full iterative structure of flow matching models, DFQL maintains higher expressivity compared to distillation-based approaches. However, the multi-step sampling process results in increased inference latency. Although our proposed action-chunking variant, QC-DFQL, partially can alleviate the execution burden by outputting multiple actions in a single inference pass, thereby reducing the required frequency of policy network execution, a more fundamental solution may lie in combining techniques such as Mean Flow Matching (Geng et al., 2025) that directly enhance inference speed at the flow network architecture level. Nevertheless, Mean Flow Matching often imposes more stringent requirements on training stability and data volume compared to standard Conditional Flow Matching. The variance introduced by the critic during RL further complicates this optimization, making the stable joint training of Mean-Flow-Matching-based policies an open and valuable challenge.

**Hyperparameter $\alpha$ in policy extraction.** Like many offline RL frameworks following the behavior-constrained paradigm (Tarasov et al., 2023b; Park et al., 2025b), the performance of DFQL is sensitive to the hyperparameter $\alpha$, which balances the BC ($\mathcal{L}_{\mathrm{BC}}$) and RL ($\mathcal{L}_{\mathrm{RL}}$) components. While some recent studies have explored adaptive strategies using dual optimization or reward-scaling heuristics, effectively automating this trade-off for generative policies remains a significant challenge. Existing mechanisms frequently suffer from high sensitivity to initial hyperparameter settings or rely on metrics such as the Kullback–Leibler divergence, which are computationally expensive to evaluate for flow matching models due to the need for ODE integration. Therefore, developing a robust, initialization-insensitive, and architecture-agnostic automated tuning mechanism for $\alpha$ represents a critical and ongoing research direction in generative policy extraction.

## C. Implementation Details and Ablation Experiments

In this section, we provide the comprehensive implementation specifics of DFQL along with ablation studies that justify our core design choices.

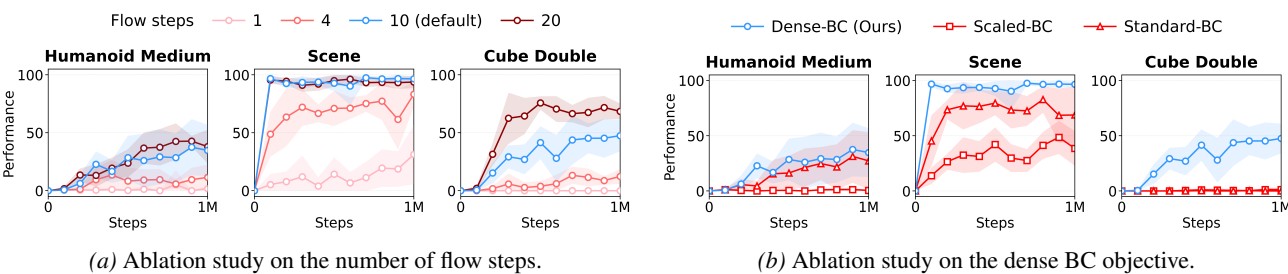

*(a)* Ablation study on the number of flow steps.            *(b)* Ablation study on the dense BC objective.

*Figure 5.* **Ablation studies on key components of our framework.**

**Discretization strategy and selection of $K$.** The generation process of our flow-based policy is implemented using Euler discretization steps without incorporating sinusoidal temporal embeddings. Theoretically, increasing the number of discrete ODE steps $K$ minimizes approximation bias, though at the expense of training and inference efficiency. As evidenced in Figure 5a, while performance generally scales with $K$, the necessity for higher values varies by task. For the majority of environments, such as "humanoidmaze-medium-singletask" and "scene-singletask", $K = 10$ is already sufficient. Although complex scenarios like "cube-double-singletask" still benefit from further increases, we prioritize the balance between performance and computational overhead. Consequently, aligning with the configuration in FQL, we adopt $K = 10$ for all other experiments and recommend it as a robust initial setting for users.

**Implementation and ablation of the dense BC objective.** Following the foundations in Section 2, our framework utilizes a flow-matching objective based on linear probability paths. To maintain structural symmetry and ensure consistent supervision density between the RL objective and the BC constraint, we adopt the dense BC objective introduced in Section 3.3. To evaluate the impact of this design (Equation 20), we conduct ablation studies across three representative tasks, comparing it against the conventional uniform random sampling scheme (results illustrated in Figure 5b). To distinguish the benefits of increased supervision density from a simple increase in constraint magnitude, we introduce two specific baselines: (i)

standard uniform temporal sampling, and (ii) a linearly scaled version where the standard BC component ($\mathcal{L}_{BC}$) is multiplied by the number of discrete steps $K$ to simulate higher intensity. Our findings reveal that the proposed dense BC objective achieves the best performance, followed by the standard approach. Notably, the linear scaling of BC intensity yields the poorest results, as it disrupts the optimal balance between BC and RL signals. This ablation highlights the utility of the dense BC design, which further bolsters our method's performance and ensures a robust training signal alongside our core framework.

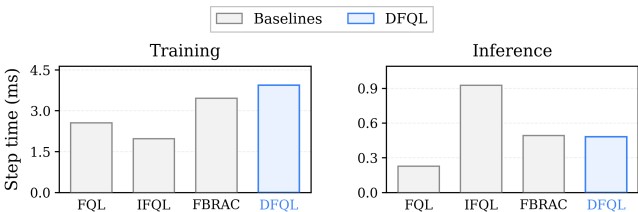

*Figure 6.* **Run time comparison.** The run times are measured on the same machine and are averaged over 8 seeds.

**Computational cost.** We profile the wall-clock cost of major flow-based baselines on the representative cube-double-play-singletask-v0 environment under the same machine, codebase, and seed setting. As shown in Figure 6, DFQL has higher inference latency than the one-step FQL baseline, while being comparable to FBRAC and faster than IFQL. This trend is consistent with the computational structures of these methods: FQL performs one-step action generation, FBRAC and DFQL both preserve an iterative flow-based sampling process, and IFQL additionally requires selecting from multiple sampled candidates. During training, DFQL incurs a higher per-update cost. This is mainly because DFQL applies RL guidance to every flow step, and the dense BC objective introduces additional step-wise supervision. Therefore, the additional runtime cost reflects the expected computational trade-off of preserving the multi-step flow structure instead of distilling it into a one-step policy.

**Value learning and target aggregation.** We maintain an ensemble of two Q-networks to enhance optimization stability. In both the policy improvement phase and the TD target calculation, the Q-values are aggregated using either the mean or the minimum, depending on the specific task environment. We also utilize target Q networks whose parameters are updated via exponential moving average. By default, the mean value is employed for both the actor's objective and the TD target. However, for Adroit and OGBench Antmaze tasks, we adopt the minimum of the two Q-values (Clipped Double Q-learning) to leverage its pessimistic advantages in these domains. These value learning configurations are strictly aligned with the settings used in FQL to ensure a fair and consistent comparison.

**Network architectures.** In alignment with the standard settings in FQL, we utilize multi-layer perceptrons with a hidden layer structure of [512, 512, 512, 512] for all neural components and incorporate layer normalization in the value networks. However, we found a rare forward-pass numerical overflow when the learned critic provides unreliable velocity targets for noise samples in extremely low-probability regions, and the actor's velocity field is iteratively integrated for multiple steps. This issue arises during multi-step sampling/integration rather than from backpropagated gradient explosion. To avoid undefined rollouts, we apply a simple numerical guardrail by bounding the actor output with $5 \times \tanh$ only in DFQL. Importantly, this modification is not part of our algorithmic contribution; it only ensures stable evaluation rollouts. As shown in Table 3, applying this bounded activation to the FQL baseline actually leads to performance degradation. Given that FQL serves as our strongest competitive baseline, we choose to maintain its original standard architecture, as well as those of other baselines, to ensure they are evaluated under their respective optimal configurations for a fair comparison.

**Online fine-tuning.** For the transition from offline to online RL, we follow a simple protocol of appending online interaction data to the existing offline buffer. The training objectives for both the actor and critic remain identical to those used in the offline phase, as described in Algorithm 1.

**Image processing.** For tasks involving pixel-based observations, we leverage a compact version of the IMPALA encoder. Specifically, we apply random-shift augmentations with a probability of 0.5 to improve generalization. Furthermore, to capture temporal dynamics effectively, we utilize frame stacking with three consecutive images as input.

**Training and evaluation.** The training duration is set to 1M gradient steps for state-based OGBench environments, while a reduced schedule of 500K steps is applied to D4RL and pixel-based OGBench tasks. Each agent undergoes evaluation every 100K steps across 50 episodes. Following the established OGBench evaluation framework (Park et al., 2025a), we

report the average success rate of the last three epochs; meanwhile, D4RL results reflect the performance at the final epoch. For offline-to-online transitions, we document performance specifically at the 1M (offline completion) and 2M (online completion) marks.

**Coefficient $\alpha$.** The most critical hyperparameter in our unified objective is the coefficient $\alpha$, which balances the imitation and reinforcement signals. We conduct a grid search for $\alpha$ over $\{0.0033, 0.01, 0.02, 0.033, 0.1, 0.2\}$ for Adroit tasks and $\{0.2, 0.33, 1, 2, 3.3, 10, 20, 33\}$ for all other environments, selecting the optimal value for each specific task. The lower range for Adroit is necessitated by the significantly larger scale of its return values compared to other benchmarks. In practice, we recommend selecting alpha through a two-stage procedure. One may first perform a coarse search to identify the appropriate order of magnitude, and then refine alpha within a small local range.

**Baselines.** We adopt the same baseline configurations as those utilized in FQL (Park et al., 2025b), ensuring full methodological alignment in both implementation and evaluation protocols. Thanks to the exhaustive and detailed experimental analysis provided by the authors of FQL, and given that our experimental setup is virtually identical to theirs, we are able to directly leverage many of their reported benchmark results for our comparative analysis to facilitate a fair assessment. All baselines are executed using the official FQL codebase without modification, with hyperparameters strictly retained as reported in the original study.

**Hyperparameters.** The complete set of hyperparameters is presented in Tables 6 to 8.

*Table 3.* **Reproduction check and bounded-activation ablation for the FQL baseline.** We compare the FQL results reported in the original paper, our reproduced results using the official FQL codebase, and a variant that adds the bounded activation function ($5 \times \tanh$) to the policy output. The reproduced results closely match the reported FQL performance, while adding the bounded activation leads to a slight decrease in overall average performance. Therefore, for all formal comparisons, we keep the standard FQL architecture without the additional bounded activation to avoid penalizing the baseline.

| Task | FQL reported | FQL reproduced | FQL + bounded tanh activation |
|---|---|---|---|
| antmaze-large-navigate-singletask-task1-v0 | $80\pm8$ | $79\pm11$ | $73\pm11$ |
| antmaze-large-navigate-singletask-task2-v0 | $57\pm10$ | $61\pm11$ | $52\pm12$ |
| antmaze-large-navigate-singletask-task3-v0 | $93\pm3$ | $95\pm4$ | $93\pm4$ |
| antmaze-large-navigate-singletask-task4-v0 | $80\pm4$ | $85\pm7$ | $79\pm10$ |
| antmaze-large-navigate-singletask-task5-v0 | $83\pm4$ | $84\pm7$ | $80\pm8$ |
| antmaze-giant-navigate-singletask-task1-v0 | $4\pm5$ | $10\pm9$ | $6\pm9$ |
| antmaze-giant-navigate-singletask-task2-v0 | $9\pm7$ | $7\pm11$ | $7\pm12$ |
| antmaze-giant-navigate-singletask-task3-v0 | $0\pm1$ | $0\pm1$ | $0\pm0$ |
| antmaze-giant-navigate-singletask-task4-v0 | $14\pm23$ | $26\pm28$ | $4\pm13$ |
| antmaze-giant-navigate-singletask-task5-v0 | $16\pm28$ | $13\pm26$ | $11\pm22$ |
| humanoidmaze-medium-navigate-singletask-task1-v0 | $19\pm12$ | $29\pm21$ | $23\pm22$ |
| humanoidmaze-medium-navigate-singletask-task2-v0 | $94\pm3$ | $96\pm6$ | $92\pm13$ |
| humanoidmaze-medium-navigate-singletask-task3-v0 | $74\pm18$ | $59\pm40$ | $53\pm37$ |
| humanoidmaze-medium-navigate-singletask-task4-v0 | $3\pm4$ | $0\pm2$ | $1\pm3$ |
| humanoidmaze-medium-navigate-singletask-task5-v0 | $97\pm2$ | $94\pm15$ | $99\pm1$ |
| humanoidmaze-large-navigate-singletask-task1-v0 | $7\pm6$ | $3\pm3$ | $4\pm6$ |
| humanoidmaze-large-navigate-singletask-task2-v0 | $0\pm0$ | $0\pm0$ | $0\pm0$ |
| humanoidmaze-large-navigate-singletask-task3-v0 | $11\pm7$ | $13\pm8$ | $11\pm11$ |
| humanoidmaze-large-navigate-singletask-task4-v0 | $2\pm3$ | $1\pm2$ | $2\pm4$ |
| humanoidmaze-large-navigate-singletask-task5-v0 | $1\pm3$ | $1\pm3$ | $3\pm6$ |
| antsoccer-arena-navigate-singletask-task1-v0 | $77\pm4$ | $79\pm7$ | $80\pm12$ |
| antsoccer-arena-navigate-singletask-task2-v0 | $88\pm3$ | $88\pm6$ | $85\pm7$ |
| antsoccer-arena-navigate-singletask-task3-v0 | $61\pm6$ | $52\pm10$ | $50\pm10$ |
| antsoccer-arena-navigate-singletask-task4-v0 | $39\pm6$ | $39\pm10$ | $36\pm10$ |
| antsoccer-arena-navigate-singletask-task5-v0 | $36\pm9$ | $32\pm14$ | $29\pm15$ |
| cube-single-play-singletask-task1-v0 | $97\pm2$ | $97\pm4$ | $97\pm3$ |
| cube-single-play-singletask-task2-v0 | $97\pm2$ | $97\pm4$ | $98\pm3$ |
| cube-single-play-singletask-task3-v0 | $98\pm2$ | $98\pm2$ | $98\pm3$ |
| cube-single-play-singletask-task4-v0 | $94\pm3$ | $95\pm3$ | $97\pm2$ |
| cube-single-play-singletask-task5-v0 | $93\pm3$ | $94\pm4$ | $93\pm6$ |
| cube-double-play-singletask-task1-v0 | $61\pm9$ | $61\pm9$ | $56\pm11$ |
| cube-double-play-singletask-task2-v0 | $36\pm6$ | $34\pm8$ | $36\pm10$ |
| cube-double-play-singletask-task3-v0 | $22\pm5$ | $20\pm7$ | $24\pm12$ |
| cube-double-play-singletask-task4-v0 | $5\pm2$ | $5\pm3$ | $5\pm2$ |
| cube-double-play-singletask-task5-v0 | $19\pm10$ | $15\pm10$ | $16\pm10$ |
| scene-play-singletask-task1-v0 | $100\pm0$ | $100\pm0$ | $100\pm1$ |
| scene-play-singletask-task2-v0 | $76\pm9$ | $83\pm13$ | $79\pm14$ |
| scene-play-singletask-task3-v0 | $98\pm1$ | $96\pm3$ | $98\pm3$ |
| scene-play-singletask-task4-v0 | $5\pm1$ | $10\pm10$ | $9\pm10$ |
| scene-play-singletask-task5-v0 | $0\pm0$ | $0\pm0$ | $0\pm0$ |
| puzzle-3x3-play-singletask-task1-v0 | $90\pm4$ | $89\pm8$ | $93\pm5$ |
| puzzle-3x3-play-singletask-task2-v0 | $16\pm5$ | $13\pm7$ | $12\pm5$ |
| puzzle-3x3-play-singletask-task3-v0 | $10\pm3$ | $9\pm4$ | $10\pm7$ |
| puzzle-3x3-play-singletask-task4-v0 | $16\pm5$ | $11\pm6$ | $13\pm7$ |
| puzzle-3x3-play-singletask-task5-v0 | $16\pm3$ | $18\pm7$ | $22\pm11$ |
| puzzle-4x4-play-singletask-task1-v0 | $34\pm8$ | $30\pm8$ | $32\pm8$ |
| puzzle-4x4-play-singletask-task2-v0 | $16\pm5$ | $14\pm5$ | $12\pm6$ |
| puzzle-4x4-play-singletask-task3-v0 | $18\pm5$ | $19\pm9$ | $22\pm5$ |
| puzzle-4x4-play-singletask-task4-v0 | $11\pm3$ | $10\pm5$ | $9\pm5$ |
| puzzle-4x4-play-singletask-task5-v0 | $7\pm3$ | $6\pm4$ | $7\pm4$ |
| **Overall Mean** | **44** | **43** | **42** |

# D. QC-DFQL

In this section, we provide a comprehensive description of QC-DFQL, a modular extension of our framework that integrates DFQL with the Q-chunking (QC) recipe. By extending flow-matching policies to a temporally extended action space, QC-DFQL leverages the stability of our proposed optimization objective while benefiting from the improved exploration and credit assignment provided by action chunking.

## D.1. Theoretical Background of Action Chunking

The core algorithmic modification of QC lies in the fundamental shift of the action dimension, which offers two-fold advantages for offline/offline-to-online RL in challenging environments. Instead of predicting a single action at each timestep,

the agent learns to output an "action chunk" $a_{n:n+h} = [a_n, a_{n+1}, \ldots, a_{n+h-1}] \in \mathbb{R}^{dh}$, which is executed open-loop for $h$ steps. The advantages of QC are mainly reflected in the following two aspects:

**(i) Temporally coherent exploration.** By operating on action sequences, the framework facilitates exploration through temporally consistent behaviors. Unlike standard Markovian policies that may produce jittery or inconsistent movements, action chunking allows the agent to leverage structured behaviors (e.g., reaching or grasping) inherent in the offline data. This consistency significantly improves state coverage and sample efficiency during the online fine-tuning phase.

**(ii) Unbiased multi-step backups.** The "chunked" formulation enables stable and accelerated value backups. While traditional $n$-step returns often suffer from off-policy bias when the behavior policy differs from the current policy, a critic trained on action chunks maintains an unbiased value estimate. This is because the $h$-step Q-function takes as input the exact sequence of actions used to collect the corresponding $h$-step rewards, allowing the value signal to propagate backward $h$ times faster without the typical bias found in uncorrected multi-step TD methods.

### D.2. Orthogonality and Synergistic Potential

A key motivation for developing QC-DFQL is the observation that QC and DFQL target distinct stages of the RL pipeline. Q-chunking primarily focuses on the representation and credit assignment stage, optimizing the structure of the action space and the efficiency of value propagation. It provides a recipe for how the agent perceives and interacts with time and action, but it does not inherently dictate how the underlying policy should be optimized.

In contrast, DFQL addresses the policy extraction and optimization stage. Its primary contribution is providing a stable optimization pathway for iterative flow-matching models by mitigating the gradient explosion problem inherent in BPTT. The original Q-chunking framework integrates the chunking design with FQL to produce a high-performance and efficient variant QC-FQL (Li et al., 2025). By replacing the FQL component with our proposed DFQL, we inherit the structural benefits of action chunking while ensuring robust optimization stability in the high-dimensional action chunk space $\mathbb{R}^{dh}$ without sacrificing the model's iterative expressive capacity. This structural synergy allows DFQL to serve as a stable backbone for the temporally extended policies required by QC, leading to superior performance in complex, sparse-reward manipulation tasks.

### D.3. Specific Training Formulas and Implementation Details

Following the established framework for temporally extended action spaces, we extract transitions consisting of consecutive state-action sequences from the dataset. Let $(s_n, \mathbf{a}_n, s_{n+h}, r_n^h) \sim \mathcal{D}$ denote a sampled transition, where $\mathbf{a}_n = [a_n, \ldots, a_{n+h-1}] \in \mathbb{R}^{dh}$ represents the action chunk and $r_n^h = \sum_{n'=0}^{h-1} \gamma^{n'} r_{n+n'}$ is the cumulative discounted reward. The optimization objectives for QC-DFQL are defined as follows:

The value function is trained to minimize the $h$-step TD error by considering the entire action sequence:

$$\mathcal{L}_Q(\phi) = \mathbb{E}_{\substack{(s_n, \mathbf{a}_n, s_{n+h}, r_n^h) \sim \mathcal{D} \\ \mathbf{a}_{n+h} \sim \pi_\theta(\cdot|s_{n+h})}} \left[ \left( Q_\phi(s_n, \mathbf{a}_n) - r_n^h - \gamma^h Q_{\bar\phi}(s_{n+h}, \mathbf{a}_{n+h}) \right)^2 \right]. \tag{40}$$

For the policy extraction stage, we optimize the velocity field $v_\theta$ by injecting gradients from the "chunked" critic. The RL guidance objective is lifted to the sequence space $\mathbb{R}^{dh}$ as:

$$\mathcal{L}_{\text{RL}}(\theta) = -\mathbb{E}_{\substack{(s_n, \mathbf{a}_n, s_{n+h}, r_n^h) \sim \mathcal{D} \\ \mathbf{a}_0^{\text{RL}} \sim \mathcal{N}(0, I_{dh})}} \left[ \sum_{k=0}^{K-1} (w_k \Delta t \cdot g^\top v_\theta(s_n, \mathbf{a}_k^{\text{RL}}, t_k)) \right], \tag{41}$$

where $g = \nabla_{\mathbf{a}_K^{\text{RL}}} Q_\phi(s_n, \mathbf{a}_K^{\text{RL}})$ and $\mathbf{a}_{k+1}^{\text{RL}} = \mathbf{a}_k^{\text{RL}} + \Delta t \cdot v_\theta(s_n, \mathbf{a}_k^{\text{RL}}, t_k)$. Simultaneously, the QC-DFQL BC loss is

$$\mathcal{L}_{\text{BC}}(\theta) = \mathbb{E}_{\substack{(s_n, \mathbf{a}_n, s_{n+h}, r_n^h) \sim \mathcal{D} \\ \mathbf{a}_0^{\text{BC}} \sim \mathcal{N}(0, I_{dh})}} \left[ \sum_{k=0}^{K-1} \|v_\theta(s_n, (1-t_k)\mathbf{a}_0^{\text{BC}} + t_k \mathbf{a}_n, t_k) - (\mathbf{a}_n - \mathbf{a}_0^{\text{BC}})\|_2^2 \right]. \tag{42}$$

By utilizing these objectives, QC-DFQL maintains optimization stability while leveraging the exploration benefits of temporally coherent actions. For the implementation, the hyperparameter $\alpha$ in QC-DFQL is set to 1 across all tasks, while all other shared configurations remain identical to those of DFQL. Regarding the chunk size, we adopt the setting $h = 5$ to maintain consistency with QC-FQL, thereby ensuring a fair comparison. The complete experimental comparison is shown in Figure 7.

# E. Full Results

The complete per-task offline RL results are provided in Table 5, while the full offline-to-online RL results are shown in Table 4. Moreover, we present the comprehensive task-by-task training curves for the QC-DFQL and QC-FQL integration study in Figure 7. We report the standard deviation using the "±" notation in the tables and indicate the 95% bootstrap confidence interval with shaded regions in the figures. Following the convention of OGBench (Park et al., 2025a), we highlight values in bold if they are at or above 95% of the best performance.

*Table 4.* **Full Offline-to-online RL results.** Results are reported as the mean over 8 seeds.

| Task | IQL | ReBRAC | Cal-QL | RLPD | IFQL | FQL | DFQL |
|---|---|---|---|---|---|---|---|
| humanoidmaze-medium-navigate | $21_{\pm13} \to 16_{\pm8}$ | $16_{\pm20} \to 1_{\pm1}$ | $0_{\pm0} \to 0_{\pm0}$ | $0_{\pm0} \to 8_{\pm10}$ | $56_{\pm35} \to \mathbf{82}_{\pm20}$ | $12_{\pm7} \to 22_{\pm12}$ | $33_{\pm26} \to 77_{\pm30}$ |
| antsoccer-arena-navigate | $2_{\pm1} \to 0_{\pm0}$ | $0_{\pm0} \to 0_{\pm0}$ | $0_{\pm0} \to 0_{\pm0}$ | $0_{\pm0} \to 0_{\pm0}$ | $26_{\pm15} \to 39_{\pm10}$ | $28_{\pm8} \to \mathbf{86}_{\pm5}$ | $47_{\pm6} \to \mathbf{89}_{\pm6}$ |
| cube-double-play | $0_{\pm1} \to 0_{\pm0}$ | $6_{\pm5} \to 28_{\pm28}$ | $0_{\pm0} \to 0_{\pm0}$ | $0_{\pm0} \to 0_{\pm0}$ | $12_{\pm9} \to 40_{\pm5}$ | $40_{\pm11} \to \mathbf{92}_{\pm3}$ | $49_{\pm23} \to \mathbf{91}_{\pm7}$ |
| scene-play | $14_{\pm11} \to 10_{\pm9}$ | $55_{\pm10} \to \mathbf{100}_{\pm0}$ | $1_{\pm2} \to 50_{\pm53}$ | $0_{\pm0} \to \mathbf{100}_{\pm0}$ | $0_{\pm1} \to 60_{\pm39}$ | $82_{\pm11} \to \mathbf{100}_{\pm1}$ | $97_{\pm5} \to \mathbf{100}_{\pm0}$ |
| puzzle-4x4-play | $5_{\pm2} \to 1_{\pm1}$ | $8_{\pm4} \to 14_{\pm35}$ | $0_{\pm0} \to 0_{\pm0}$ | $0_{\pm0} \to \mathbf{100}_{\pm1}$ | $23_{\pm6} \to 19_{\pm33}$ | $38_{\pm3} \to 38_{\pm52}$ | $14_{\pm3} \to 87_{\pm35}$ |
| antmaze-umaze-v2 | $77 \to \mathbf{96}$ | $98 \to 75$ | $77 \to \mathbf{100}$ | $0_{\pm0} \to 98_{\pm3}$ | $94_{\pm5} \to \mathbf{96}_{\pm2}$ | $97_{\pm2} \to 99_{\pm1}$ | $95_{\pm3} \to \mathbf{100}_{\pm0}$ |
| antmaze-umaze-diverse-v2 | $60 \to 64$ | $74 \to \mathbf{98}$ | $32 \to \mathbf{98}$ | $0_{\pm0} \to 94_{\pm5}$ | $69_{\pm20} \to 93_{\pm5}$ | $79_{\pm16} \to \mathbf{100}_{\pm1}$ | $90_{\pm4} \to \mathbf{99}_{\pm2}$ |
| antmaze-medium-play-v2 | $72 \to 90$ | $88 \to \mathbf{98}$ | $72 \to \mathbf{99}$ | $0_{\pm0} \to 98_{\pm2}$ | $52_{\pm19} \to 93_{\pm2}$ | $77_{\pm7} \to \mathbf{97}_{\pm2}$ | $84_{\pm5} \to \mathbf{96}_{\pm4}$ |
| antmaze-medium-diverse-v2 | $64 \to 92$ | $85 \to \mathbf{99}$ | $62 \to \mathbf{98}$ | $0_{\pm0} \to 97_{\pm2}$ | $44_{\pm26} \to 89_{\pm4}$ | $55_{\pm19} \to \mathbf{97}_{\pm3}$ | $79_{\pm8} \to \mathbf{98}_{\pm2}$ |
| antmaze-large-play-v2 | $38 \to 64$ | $68 \to 32$ | $32 \to \mathbf{97}$ | $0_{\pm0} \to 93_{\pm5}$ | $64_{\pm14} \to 80_{\pm5}$ | $66_{\pm40} \to 84_{\pm30}$ | $75_{\pm5} \to 92_{\pm3}$ |
| antmaze-large-diverse-v2 | $27 \to 64$ | $67 \to 72$ | $44 \to \mathbf{92}$ | $0_{\pm0} \to 94_{\pm3}$ | $69_{\pm6} \to 86_{\pm5}$ | $75_{\pm14} \to \mathbf{94}_{\pm3}$ | $75_{\pm7} \to \mathbf{93}_{\pm4}$ |
| pen-cloned-v1 | $84 \to 102$ | $74 \to 138$ | $-3 \to -3$ | $3_{\pm2} \to 120_{\pm10}$ | $77_{\pm7} \to 107_{\pm10}$ | $53_{\pm14} \to \mathbf{149}_{\pm6}$ | $69_{\pm7} \to \mathbf{145}_{\pm5}$ |
| door-cloned-v1 | $1 \to 20$ | $0 \to \mathbf{102}$ | $-0 \to -0$ | $0_{\pm0} \to 102_{\pm7}$ | $3_{\pm2} \to 50_{\pm15}$ | $0_{\pm0} \to \mathbf{102}_{\pm5}$ | $0_{\pm0} \to 95_{\pm2}$ |
| hammer-cloned-v1 | $1 \to 57$ | $7 \to 125$ | $0 \to 0$ | $0_{\pm0} \to 128_{\pm29}$ | $4_{\pm2} \to 60_{\pm14}$ | $0_{\pm0} \to 127_{\pm17}$ | $0_{\pm0} \to \mathbf{134}_{\pm4}$ |
| relocate-cloned-v1 | $0 \to 0$ | $1 \to 7$ | $-0 \to -0$ | $0_{\pm0} \to 2_{\pm2}$ | $-0_{\pm0} \to 5_{\pm3}$ | $0_{\pm1} \to \mathbf{62}_{\pm8}$ | $0_{\pm0} \to 49_{\pm14}$ |

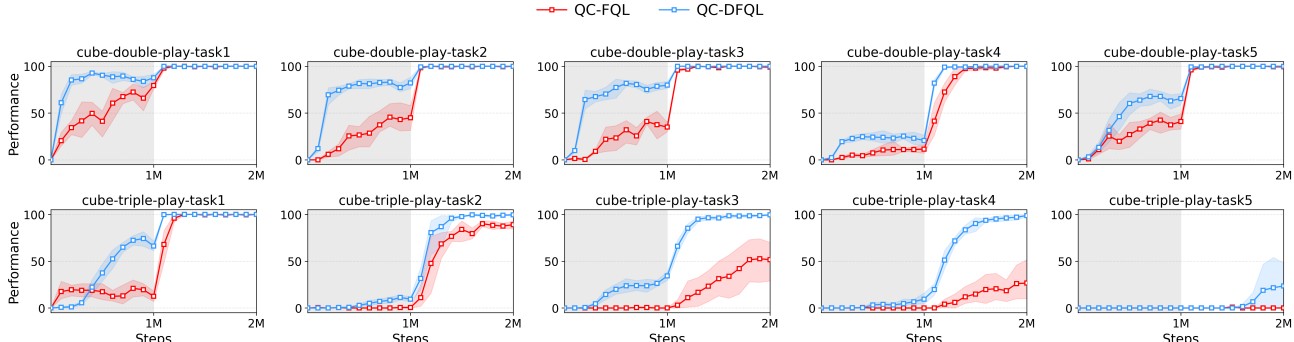

*Figure 7.* **Full QC-DFQL and QC-FQL results.** Experimental results show that QC-DFQL outperforms QC-FQL on nearly all tasks, indicating that DFQL, like FQL, can serve as a flow-matching-based module in offline/offline-to-online RL and can be integrated with other components to further improve performance.

*Table 5.* **Full offline RL results.** We report comprehensive offline RL results across all 73 tasks from the OGBench and D4RL suites. The default task for each environment is marked with an asterisk (∗).

| Task category | Gaussian policies | | | Diffusion policies | | | Flow policies | | | | |
|---|---|---|---|---|---|---|---|---|---|---|---|
| | BC | IQL | ReBRAC | IDQL | SRPO | CAC | FAWAC | FBRAC | IFQL | FQL | DFQL |
| antmaze-large-navigate-singletask-task1-v0 (∗) | $0_{\pm 0}$ | $48_{\pm 9}$ | $\mathbf{91}_{\pm 10}$ | $0_{\pm 0}$ | $0_{\pm 0}$ | $42_{\pm 7}$ | $1_{\pm 1}$ | $70_{\pm 20}$ | $24_{\pm 17}$ | $80_{\pm 8}$ | $\mathbf{93}_{\pm 4}$ |
| antmaze-large-navigate-singletask-task2-v0 | $6_{\pm 3}$ | $42_{\pm 6}$ | $\mathbf{88}_{\pm 4}$ | $14_{\pm 8}$ | $4_{\pm 4}$ | $1_{\pm 1}$ | $0_{\pm 1}$ | $35_{\pm 12}$ | $8_{\pm 3}$ | $57_{\pm 10}$ | $70_{\pm 9}$ |
| antmaze-large-navigate-singletask-task3-v0 | $29_{\pm 5}$ | $72_{\pm 7}$ | $51_{\pm 18}$ | $26_{\pm 8}$ | $3_{\pm 2}$ | $49_{\pm 10}$ | $12_{\pm 4}$ | $83_{\pm 15}$ | $52_{\pm 17}$ | $93_{\pm 3}$ | $\mathbf{97}_{\pm 3}$ |
| antmaze-large-navigate-singletask-task4-v0 | $8_{\pm 3}$ | $51_{\pm 9}$ | $84_{\pm 7}$ | $62_{\pm 25}$ | $45_{\pm 19}$ | $17_{\pm 6}$ | $10_{\pm 3}$ | $37_{\pm 18}$ | $18_{\pm 8}$ | $80_{\pm 4}$ | $\mathbf{89}_{\pm 4}$ |
| antmaze-large-navigate-singletask-task5-v0 | $10_{\pm 3}$ | $54_{\pm 22}$ | $\mathbf{90}_{\pm 2}$ | $2_{\pm 2}$ | $1_{\pm 1}$ | $55_{\pm 6}$ | $9_{\pm 5}$ | $76_{\pm 8}$ | $38_{\pm 18}$ | $83_{\pm 4}$ | $\mathbf{90}_{\pm 4}$ |
| antmaze-giant-navigate-singletask-task1-v0 (∗) | $0_{\pm 0}$ | $0_{\pm 0}$ | $\mathbf{27}_{\pm 22}$ | $0_{\pm 0}$ | $0_{\pm 0}$ | $0_{\pm 0}$ | $0_{\pm 0}$ | $0_{\pm 1}$ | $0_{\pm 0}$ | $4_{\pm 5}$ | $11_{\pm 7}$ |
| antmaze-giant-navigate-singletask-task2-v0 | $0_{\pm 0}$ | $1_{\pm 1}$ | $16_{\pm 17}$ | $0_{\pm 0}$ | $0_{\pm 0}$ | $0_{\pm 0}$ | $0_{\pm 0}$ | $4_{\pm 7}$ | $0_{\pm 0}$ | $9_{\pm 7}$ | $\mathbf{48}_{\pm 13}$ |
| antmaze-giant-navigate-singletask-task3-v0 | $0_{\pm 0}$ | $0_{\pm 0}$ | $\mathbf{34}_{\pm 22}$ | $0_{\pm 0}$ | $0_{\pm 0}$ | $0_{\pm 0}$ | $0_{\pm 0}$ | $0_{\pm 0}$ | $0_{\pm 0}$ | $0_{\pm 1}$ | $4_{\pm 4}$ |
| antmaze-giant-navigate-singletask-task4-v0 | $0_{\pm 0}$ | $0_{\pm 0}$ | $5_{\pm 12}$ | $0_{\pm 0}$ | $0_{\pm 0}$ | $0_{\pm 0}$ | $0_{\pm 0}$ | $9_{\pm 4}$ | $0_{\pm 0}$ | $14_{\pm 23}$ | $\mathbf{56}_{\pm 11}$ |
| antmaze-giant-navigate-singletask-task5-v0 | $1_{\pm 1}$ | $19_{\pm 7}$ | $49_{\pm 22}$ | $0_{\pm 1}$ | $0_{\pm 0}$ | $0_{\pm 0}$ | $0_{\pm 0}$ | $6_{\pm 10}$ | $13_{\pm 9}$ | $16_{\pm 28}$ | $\mathbf{77}_{\pm 7}$ |
| humanoidmaze-medium-navigate-singletask-task1-v0 (∗) | $1_{\pm 0}$ | $32_{\pm 7}$ | $16_{\pm 9}$ | $1_{\pm 1}$ | $0_{\pm 0}$ | $38_{\pm 19}$ | $6_{\pm 2}$ | $25_{\pm 8}$ | $69_{\pm 19}$ | $19_{\pm 12}$ | $35_{\pm 25}$ |
| humanoidmaze-medium-navigate-singletask-task2-v0 | $1_{\pm 0}$ | $41_{\pm 9}$ | $18_{\pm 16}$ | $1_{\pm 1}$ | $1_{\pm 1}$ | $47_{\pm 35}$ | $40_{\pm 2}$ | $76_{\pm 10}$ | $85_{\pm 11}$ | $\mathbf{94}_{\pm 3}$ | $93_{\pm 4}$ |
| humanoidmaze-medium-navigate-singletask-task3-v0 | $6_{\pm 2}$ | $25_{\pm 5}$ | $36_{\pm 13}$ | $0_{\pm 1}$ | $2_{\pm 1}$ | $\mathbf{83}_{\pm 18}$ | $19_{\pm 2}$ | $27_{\pm 11}$ | $49_{\pm 49}$ | $74_{\pm 18}$ | $76_{\pm 31}$ |
| humanoidmaze-medium-navigate-singletask-task4-v0 | $0_{\pm 0}$ | $0_{\pm 1}$ | $15_{\pm 16}$ | $1_{\pm 1}$ | $1_{\pm 1}$ | $5_{\pm 4}$ | $1_{\pm 1}$ | $1_{\pm 2}$ | $1_{\pm 1}$ | $3_{\pm 4}$ | $\mathbf{32}_{\pm 19}$ |
| humanoidmaze-medium-navigate-singletask-task5-v0 | $2_{\pm 1}$ | $66_{\pm 4}$ | $24_{\pm 20}$ | $1_{\pm 1}$ | $3_{\pm 3}$ | $91_{\pm 5}$ | $31_{\pm 7}$ | $63_{\pm 9}$ | $\mathbf{98}_{\pm 2}$ | $97_{\pm 2}$ | $93_{\pm 12}$ |
| humanoidmaze-large-navigate-singletask-task1-v0 (∗) | $0_{\pm 0}$ | $3_{\pm 1}$ | $2_{\pm 1}$ | $0_{\pm 0}$ | $0_{\pm 0}$ | $1_{\pm 1}$ | $0_{\pm 0}$ | $0_{\pm 0}$ | $6_{\pm 2}$ | $7_{\pm 6}$ | $\mathbf{10}_{\pm 9}$ |
| humanoidmaze-large-navigate-singletask-task2-v0 | $\mathbf{0}_{\pm 0}$ | $\mathbf{0}_{\pm 0}$ | $\mathbf{0}_{\pm 0}$ | $\mathbf{0}_{\pm 0}$ | $\mathbf{0}_{\pm 0}$ | $\mathbf{0}_{\pm 0}$ | $\mathbf{0}_{\pm 0}$ | $\mathbf{0}_{\pm 0}$ | $\mathbf{0}_{\pm 0}$ | $\mathbf{0}_{\pm 0}$ | $\mathbf{0}_{\pm 0}$ |
| humanoidmaze-large-navigate-singletask-task3-v0 | $1_{\pm 1}$ | $7_{\pm 3}$ | $8_{\pm 4}$ | $3_{\pm 1}$ | $1_{\pm 1}$ | $2_{\pm 3}$ | $1_{\pm 1}$ | $10_{\pm 2}$ | $\mathbf{48}_{\pm 10}$ | $11_{\pm 7}$ | $20_{\pm 10}$ |
| humanoidmaze-large-navigate-singletask-task4-v0 | $1_{\pm 0}$ | $1_{\pm 0}$ | $1_{\pm 1}$ | $0_{\pm 0}$ | $0_{\pm 0}$ | $0_{\pm 1}$ | $0_{\pm 0}$ | $0_{\pm 0}$ | $1_{\pm 1}$ | $\mathbf{2}_{\pm 3}$ | $1_{\pm 2}$ |
| humanoidmaze-large-navigate-singletask-task5-v0 | $0_{\pm 1}$ | $1_{\pm 1}$ | $2_{\pm 2}$ | $0_{\pm 0}$ | $0_{\pm 0}$ | $0_{\pm 0}$ | $0_{\pm 0}$ | $1_{\pm 1}$ | $0_{\pm 0}$ | $1_{\pm 3}$ | $\mathbf{7}_{\pm 6}$ |
| antsoccer-arena-navigate-singletask-task1-v0 | $2_{\pm 1}$ | $14_{\pm 5}$ | $0_{\pm 0}$ | $44_{\pm 12}$ | $2_{\pm 1}$ | $1_{\pm 3}$ | $22_{\pm 2}$ | $17_{\pm 3}$ | $61_{\pm 25}$ | $\mathbf{77}_{\pm 4}$ | $\mathbf{77}_{\pm 7}$ |
| antsoccer-arena-navigate-singletask-task2-v0 | $2_{\pm 2}$ | $17_{\pm 7}$ | $0_{\pm 1}$ | $15_{\pm 12}$ | $3_{\pm 1}$ | $0_{\pm 0}$ | $8_{\pm 1}$ | $82_{\pm 3}$ | $75_{\pm 3}$ | $\mathbf{88}_{\pm 3}$ | $\mathbf{88}_{\pm 6}$ |
| antsoccer-arena-navigate-singletask-task3-v0 | $0_{\pm 0}$ | $6_{\pm 4}$ | $0_{\pm 0}$ | $0_{\pm 0}$ | $0_{\pm 0}$ | $8_{\pm 19}$ | $11_{\pm 5}$ | $16_{\pm 3}$ | $14_{\pm 22}$ | $\mathbf{61}_{\pm 6}$ | $51_{\pm 10}$ |
| antsoccer-arena-navigate-singletask-task4-v0 (∗) | $1_{\pm 0}$ | $3_{\pm 2}$ | $0_{\pm 0}$ | $0_{\pm 1}$ | $0_{\pm 0}$ | $0_{\pm 0}$ | $12_{\pm 3}$ | $24_{\pm 4}$ | $16_{\pm 9}$ | $\mathbf{39}_{\pm 6}$ | $40_{\pm 7}$ |
| antsoccer-arena-navigate-singletask-task5-v0 | $0_{\pm 0}$ | $2_{\pm 2}$ | $0_{\pm 0}$ | $0_{\pm 0}$ | $0_{\pm 0}$ | $0_{\pm 0}$ | $9_{\pm 2}$ | $15_{\pm 4}$ | $0_{\pm 1}$ | $36_{\pm 9}$ | $\mathbf{38}_{\pm 13}$ |
| cube-single-play-singletask-task1-v0 | $10_{\pm 5}$ | $88_{\pm 3}$ | $89_{\pm 5}$ | $95_{\pm 2}$ | $89_{\pm 7}$ | $77_{\pm 28}$ | $81_{\pm 9}$ | $73_{\pm 33}$ | $79_{\pm 4}$ | $\mathbf{97}_{\pm 2}$ | $\mathbf{97}_{\pm 2}$ |
| cube-single-play-singletask-task2-v0 (∗) | $3_{\pm 1}$ | $85_{\pm 8}$ | $92_{\pm 4}$ | $96_{\pm 2}$ | $82_{\pm 16}$ | $80_{\pm 30}$ | $81_{\pm 9}$ | $83_{\pm 13}$ | $73_{\pm 3}$ | $97_{\pm 2}$ | $\mathbf{99}_{\pm 2}$ |
| cube-single-play-singletask-task3-v0 | $9_{\pm 3}$ | $91_{\pm 5}$ | $93_{\pm 3}$ | $99_{\pm 1}$ | $96_{\pm 2}$ | $98_{\pm 1}$ | $87_{\pm 4}$ | $82_{\pm 12}$ | $88_{\pm 4}$ | $98_{\pm 2}$ | $\mathbf{100}_{\pm 1}$ |
| cube-single-play-singletask-task4-v0 | $2_{\pm 1}$ | $73_{\pm 6}$ | $\mathbf{92}_{\pm 3}$ | $93_{\pm 4}$ | $70_{\pm 18}$ | $91_{\pm 2}$ | $79_{\pm 6}$ | $79_{\pm 20}$ | $79_{\pm 6}$ | $\mathbf{94}_{\pm 3}$ | $\mathbf{94}_{\pm 4}$ |
| cube-single-play-singletask-task5-v0 | $3_{\pm 3}$ | $78_{\pm 9}$ | $87_{\pm 8}$ | $90_{\pm 6}$ | $61_{\pm 12}$ | $80_{\pm 20}$ | $78_{\pm 10}$ | $76_{\pm 33}$ | $77_{\pm 7}$ | $\mathbf{93}_{\pm 3}$ | $91_{\pm 5}$ |
| cube-double-play-singletask-task1-v0 | $8_{\pm 3}$ | $27_{\pm 5}$ | $45_{\pm 6}$ | $39_{\pm 19}$ | $7_{\pm 6}$ | $21_{\pm 8}$ | $21_{\pm 7}$ | $47_{\pm 11}$ | $35_{\pm 9}$ | $\mathbf{61}_{\pm 9}$ | $62_{\pm 20}$ |
| cube-double-play-singletask-task2-v0 (∗) | $0_{\pm 0}$ | $1_{\pm 1}$ | $7_{\pm 3}$ | $16_{\pm 10}$ | $0_{\pm 0}$ | $2_{\pm 2}$ | $2_{\pm 1}$ | $22_{\pm 12}$ | $9_{\pm 5}$ | $36_{\pm 6}$ | $\mathbf{46}_{\pm 17}$ |
| cube-double-play-singletask-task3-v0 | $0_{\pm 0}$ | $0_{\pm 0}$ | $4_{\pm 1}$ | $17_{\pm 8}$ | $0_{\pm 1}$ | $3_{\pm 1}$ | $1_{\pm 1}$ | $4_{\pm 2}$ | $8_{\pm 5}$ | $22_{\pm 5}$ | $\mathbf{46}_{\pm 16}$ |
| cube-double-play-singletask-task4-v0 | $0_{\pm 0}$ | $0_{\pm 0}$ | $1_{\pm 1}$ | $0_{\pm 1}$ | $0_{\pm 0}$ | $0_{\pm 1}$ | $0_{\pm 0}$ | $0_{\pm 1}$ | $1_{\pm 1}$ | $5_{\pm 2}$ | $\mathbf{9}_{\pm 6}$ |
| cube-double-play-singletask-task5-v0 | $0_{\pm 0}$ | $4_{\pm 3}$ | $4_{\pm 2}$ | $1_{\pm 1}$ | $0_{\pm 0}$ | $3_{\pm 2}$ | $2_{\pm 1}$ | $2_{\pm 2}$ | $17_{\pm 6}$ | $19_{\pm 10}$ | $\mathbf{27}_{\pm 20}$ |
| scene-play-singletask-task1-v0 | $19_{\pm 6}$ | $94_{\pm 3}$ | $95_{\pm 2}$ | $\mathbf{100}_{\pm 0}$ | $94_{\pm 4}$ | $\mathbf{100}_{\pm 1}$ | $87_{\pm 8}$ | $96_{\pm 8}$ | $98_{\pm 3}$ | $\mathbf{100}_{\pm 0}$ | $\mathbf{100}_{\pm 0}$ |
| scene-play-singletask-task2-v0 (∗) | $1_{\pm 1}$ | $12_{\pm 3}$ | $50_{\pm 13}$ | $33_{\pm 14}$ | $2_{\pm 2}$ | $50_{\pm 40}$ | $18_{\pm 8}$ | $46_{\pm 10}$ | $0_{\pm 0}$ | $76_{\pm 9}$ | $\mathbf{97}_{\pm 3}$ |
| scene-play-singletask-task3-v0 | $1_{\pm 1}$ | $32_{\pm 7}$ | $55_{\pm 16}$ | $94_{\pm 4}$ | $4_{\pm 4}$ | $49_{\pm 16}$ | $38_{\pm 9}$ | $78_{\pm 14}$ | $54_{\pm 19}$ | $\mathbf{98}_{\pm 1}$ | $\mathbf{98}_{\pm 2}$ |
| scene-play-singletask-task4-v0 | $2_{\pm 2}$ | $0_{\pm 1}$ | $3_{\pm 3}$ | $4_{\pm 3}$ | $0_{\pm 0}$ | $0_{\pm 0}$ | $6_{\pm 1}$ | $4_{\pm 4}$ | $0_{\pm 0}$ | $5_{\pm 1}$ | $\mathbf{9}_{\pm 13}$ |
| scene-play-singletask-task5-v0 | $\mathbf{0}_{\pm 0}$ | $\mathbf{0}_{\pm 0}$ | $\mathbf{0}_{\pm 0}$ | $\mathbf{0}_{\pm 0}$ | $\mathbf{0}_{\pm 0}$ | $\mathbf{0}_{\pm 0}$ | $\mathbf{0}_{\pm 0}$ | $\mathbf{0}_{\pm 0}$ | $\mathbf{0}_{\pm 0}$ | $\mathbf{0}_{\pm 0}$ | $\mathbf{0}_{\pm 0}$ |
| puzzle-3x3-play-singletask-task1-v0 | $5_{\pm 2}$ | $33_{\pm 6}$ | $\mathbf{97}_{\pm 4}$ | $52_{\pm 12}$ | $89_{\pm 5}$ | $\mathbf{97}_{\pm 2}$ | $25_{\pm 9}$ | $63_{\pm 19}$ | $\mathbf{94}_{\pm 3}$ | $90_{\pm 4}$ | $\mathbf{94}_{\pm 8}$ |
| puzzle-3x3-play-singletask-task2-v0 | $1_{\pm 1}$ | $4_{\pm 3}$ | $1_{\pm 1}$ | $0_{\pm 1}$ | $0_{\pm 1}$ | $0_{\pm 0}$ | $4_{\pm 2}$ | $2_{\pm 2}$ | $1_{\pm 2}$ | $16_{\pm 5}$ | $\mathbf{44}_{\pm 26}$ |
| puzzle-3x3-play-singletask-task3-v0 | $1_{\pm 1}$ | $3_{\pm 2}$ | $3_{\pm 1}$ | $0_{\pm 0}$ | $0_{\pm 0}$ | $0_{\pm 0}$ | $1_{\pm 0}$ | $1_{\pm 1}$ | $0_{\pm 0}$ | $10_{\pm 3}$ | $\mathbf{33}_{\pm 13}$ |
| puzzle-3x3-play-singletask-task4-v0 (∗) | $1_{\pm 1}$ | $2_{\pm 1}$ | $2_{\pm 1}$ | $0_{\pm 0}$ | $0_{\pm 0}$ | $0_{\pm 0}$ | $1_{\pm 1}$ | $2_{\pm 2}$ | $0_{\pm 0}$ | $16_{\pm 5}$ | $\mathbf{81}_{\pm 9}$ |
| puzzle-3x3-play-singletask-task5-v0 | $1_{\pm 0}$ | $3_{\pm 2}$ | $5_{\pm 3}$ | $0_{\pm 0}$ | $0_{\pm 0}$ | $0_{\pm 0}$ | $1_{\pm 1}$ | $2_{\pm 2}$ | $0_{\pm 0}$ | $16_{\pm 3}$ | $\mathbf{75}_{\pm 20}$ |
| puzzle-4x4-play-singletask-task1-v0 | $1_{\pm 1}$ | $12_{\pm 2}$ | $26_{\pm 4}$ | $48_{\pm 5}$ | $24_{\pm 9}$ | $44_{\pm 10}$ | $1_{\pm 2}$ | $32_{\pm 9}$ | $49_{\pm 9}$ | $34_{\pm 8}$ | $\mathbf{53}_{\pm 8}$ |
| puzzle-4x4-play-singletask-task2-v0 | $0_{\pm 0}$ | $7_{\pm 4}$ | $12_{\pm 4}$ | $14_{\pm 5}$ | $0_{\pm 1}$ | $0_{\pm 0}$ | $0_{\pm 1}$ | $5_{\pm 3}$ | $4_{\pm 4}$ | $16_{\pm 5}$ | $\mathbf{17}_{\pm 6}$ |
| puzzle-4x4-play-singletask-task3-v0 | $0_{\pm 0}$ | $9_{\pm 3}$ | $15_{\pm 3}$ | $34_{\pm 5}$ | $21_{\pm 10}$ | $29_{\pm 12}$ | $1_{\pm 1}$ | $20_{\pm 10}$ | $\mathbf{50}_{\pm 14}$ | $18_{\pm 5}$ | $33_{\pm 8}$ |
| puzzle-4x4-play-singletask-task4-v0 (∗) | $0_{\pm 0}$ | $5_{\pm 2}$ | $10_{\pm 3}$ | $\mathbf{26}_{\pm 6}$ | $7_{\pm 4}$ | $1_{\pm 1}$ | $0_{\pm 0}$ | $5_{\pm 1}$ | $21_{\pm 11}$ | $11_{\pm 3}$ | $14_{\pm 6}$ |
| puzzle-4x4-play-singletask-task5-v0 | $0_{\pm 0}$ | $4_{\pm 1}$ | $7_{\pm 3}$ | $\mathbf{24}_{\pm 11}$ | $1_{\pm 1}$ | $0_{\pm 0}$ | $0_{\pm 1}$ | $4_{\pm 3}$ | $2_{\pm 2}$ | $7_{\pm 3}$ | $9_{\pm 4}$ |
| antmaze-umaze-v2 | $55$ | $77$ | $\mathbf{98}$ | $94$ | $97$ | $66_{\pm 5}$ | $90_{\pm 6}$ | $\mathbf{94}_{\pm 3}$ | $92_{\pm 6}$ | $\mathbf{96}_{\pm 2}$ | $\mathbf{96}_{\pm 3}$ |
| antmaze-umaze-diverse-v2 | $47$ | $54$ | $84$ | $80$ | $82$ | $66_{\pm 11}$ | $55_{\pm 7}$ | $82_{\pm 9}$ | $62_{\pm 12}$ | $\mathbf{89}_{\pm 5}$ | $86_{\pm 7}$ |
| antmaze-medium-play-v2 | $0$ | $66$ | $\mathbf{90}$ | $84$ | $81$ | $49_{\pm 24}$ | $52_{\pm 12}$ | $77_{\pm 7}$ | $56_{\pm 15}$ | $78_{\pm 7}$ | $81_{\pm 7}$ |
| antmaze-medium-diverse-v2 | $1$ | $74$ | $84$ | $85$ | $75$ | $0_{\pm 1}$ | $44_{\pm 15}$ | $77_{\pm 6}$ | $60_{\pm 25}$ | $71_{\pm 13}$ | $77_{\pm 6}$ |
| antmaze-large-play-v2 | $0$ | $42$ | $52$ | $64$ | $54$ | $0_{\pm 0}$ | $10_{\pm 6}$ | $32_{\pm 21}$ | $55_{\pm 9}$ | $\mathbf{84}_{\pm 7}$ | $76_{\pm 7}$ |
| antmaze-large-diverse-v2 | $0$ | $30$ | $64$ | $68$ | $54$ | $0_{\pm 0}$ | $16_{\pm 10}$ | $20_{\pm 17}$ | $64_{\pm 8}$ | $\mathbf{83}_{\pm 4}$ | $77_{\pm 6}$ |
| pen-human-v1 | $71$ | $78$ | $\mathbf{103}$ | $76_{\pm 10}$ | $69_{\pm 7}$ | $64_{\pm 8}$ | $67_{\pm 5}$ | $77_{\pm 7}$ | $71_{\pm 12}$ | $53_{\pm 6}$ | $71_{\pm 13}$ |
| pen-cloned-v1 | $52$ | $83$ | $\mathbf{103}$ | $64_{\pm 7}$ | $61_{\pm 7}$ | $56_{\pm 10}$ | $62_{\pm 10}$ | $67_{\pm 9}$ | $80_{\pm 11}$ | $74_{\pm 11}$ | $68_{\pm 4}$ |
| pen-expert-v1 | $110$ | $128$ | $152$ | $140_{\pm 6}$ | $134_{\pm 4}$ | $103_{\pm 9}$ | $118_{\pm 6}$ | $119_{\pm 7}$ | $139_{\pm 5}$ | $142_{\pm 6}$ | $140_{\pm 9}$ |
| door-human-v1 | $2$ | $3$ | $-0$ | $6_{\pm 2}$ | $3_{\pm 3}$ | $5_{\pm 2}$ | $2_{\pm 1}$ | $4_{\pm 2}$ | $\mathbf{7}_{\pm 2}$ | $0_{\pm 0}$ | $0_{\pm 1}$ |
| door-cloned-v1 | $-0$ | $\mathbf{3}$ | $0$ | $0_{\pm 0}$ | $0_{\pm 0}$ | $1_{\pm 0}$ | $0_{\pm 1}$ | $0_{\pm 0}$ | $2_{\pm 2}$ | $2_{\pm 1}$ | $0_{\pm 0}$ |
| door-expert-v1 | $\mathbf{105}$ | $107$ | $106$ | $105_{\pm 1}$ | $105_{\pm 0}$ | $98_{\pm 3}$ | $103_{\pm 1}$ | $104_{\pm 1}$ | $104_{\pm 2}$ | $104_{\pm 1}$ | $104_{\pm 1}$ |
| hammer-human-v1 | $3$ | $2$ | $0$ | $2_{\pm 1}$ | $1_{\pm 1}$ | $2_{\pm 0}$ | $2_{\pm 1}$ | $2_{\pm 1}$ | $\mathbf{3}_{\pm 1}$ | $1_{\pm 1}$ | $0_{\pm 0}$ |
| hammer-cloned-v1 | $1$ | $2$ | $5$ | $2_{\pm 1}$ | $2_{\pm 1}$ | $1_{\pm 1}$ | $1_{\pm 0}$ | $2_{\pm 1}$ | $2_{\pm 1}$ | $\mathbf{11}_{\pm 9}$ | $6_{\pm 3}$ |
| hammer-expert-v1 | $127$ | $\mathbf{129}$ | $134$ | $125_{\pm 4}$ | $127_{\pm 0}$ | $92_{\pm 11}$ | $118_{\pm 3}$ | $119_{\pm 4}$ | $117_{\pm 9}$ | $125_{\pm 3}$ | $123_{\pm 3}$ |
| relocate-human-v1 | $\mathbf{0}$ | $\mathbf{0}$ | $\mathbf{0}$ | $\mathbf{0}_{\pm 0}$ | $\mathbf{0}_{\pm 0}$ | $\mathbf{0}_{\pm 0}$ | $\mathbf{0}_{\pm 0}$ | $\mathbf{0}_{\pm 0}$ | $\mathbf{0}_{\pm 0}$ | $\mathbf{0}_{\pm 0}$ | $\mathbf{0}_{\pm 0}$ |
| relocate-cloned-v1 | $-0$ | $0$ | $2$ | $-0_{\pm 0}$ | $-0_{\pm 0}$ | $-0_{\pm 0}$ | $-0_{\pm 0}$ | $1_{\pm 1}$ | $-0_{\pm 0}$ | $-0_{\pm 0}$ | $0_{\pm 0}$ |
| relocate-expert-v1 | $\mathbf{108}$ | $106$ | $108$ | $107_{\pm 1}$ | $106_{\pm 2}$ | $93_{\pm 6}$ | $105_{\pm 3}$ | $105_{\pm 2}$ | $104_{\pm 3}$ | $107_{\pm 1}$ | $\mathbf{108}_{\pm 2}$ |
| visual-cube-single-play-singletask-task1-v0 | — | $70_{\pm 12}$ | $83_{\pm 6}$ | — | — | — | — | $55_{\pm 8}$ | $49_{\pm 7}$ | $81_{\pm 12}$ | $\mathbf{96}_{\pm 3}$ |
| visual-cube-double-play-singletask-task1-v0 | — | $\mathbf{34}_{\pm 23}$ | $4_{\pm 4}$ | — | — | — | — | $6_{\pm 2}$ | $8_{\pm 6}$ | $21_{\pm 11}$ | $27_{\pm 21}$ |
| visual-scene-play-singletask-task1-v0 | — | $\mathbf{97}_{\pm 2}$ | $98_{\pm 4}$ | — | — | — | — | $46_{\pm 4}$ | $86_{\pm 10}$ | $98_{\pm 3}$ | $\mathbf{100}_{\pm 1}$ |
| visual-puzzle-3x3-play-singletask-task1-v0 | — | $7_{\pm 15}$ | $88_{\pm 4}$ | — | — | — | — | $7_{\pm 2}$ | $\mathbf{100}_{\pm 0}$ | $94_{\pm 1}$ | $99_{\pm 2}$ |
| visual-puzzle-4x4-play-singletask-task1-v0 | — | $0_{\pm 0}$ | $26_{\pm 6}$ | — | — | — | — | $0_{\pm 0}$ | $8_{\pm 15}$ | $33_{\pm 6}$ | $\mathbf{40}_{\pm 9}$ |

# F. Experimental Details

## F.1. Environments

Our implementation of DFQL and baselines is based on the JAX framework (Bradbury et al., 2018) and builds on the official OGBench reference implementations.

**OGBench (Park et al., 2025a):**
Our main evaluation suite is OGBench. We select 10 environments, covering 50 state-based tasks and 5 pixel-based tasks. Since OGBench is primarily designed for goal-conditioned offline RL, we follow its *single-task* setting (environments with the "-singletask" suffix) to benchmark standard reward-maximizing offline RL methods.

In OGBench, each environment includes five evaluation goals, corresponding to five tasks ("-singletask-task1" to "-singletask-task5"), with "-singletask" (no suffix) denoting the default task. We adopt the single-task reward/termination specification from the benchmark (Park et al., 2025a). Concretely, rewards are semi-sparse and depend on goal progress: for locomotion-style tasks that typically contain a single subtask (e.g., "reach the goal"), the reward is either $-1$ or $0$; for manipulation-style tasks with multiple subtasks (e.g., "open the drawer", "turn the first button's color blue"), rewards lie in $[-n_{\text{task}}, 0]$, where $n_{\text{task}}$ is the number of subtasks (up to 16 in our selected environments). Episodes terminate upon completing the final goal, consistent with the official protocol (Park et al., 2025a).

We evaluate the following datasets (five tasks per environment), resulting in ten state-based datasets and five pixel-based datasets:

- State-based datasets:
    - antmaze-large-navigate-v0
    - antmaze-giant-navigate-v0
    - humanoidmaze-medium-navigate-v0
    - humanoidmaze-large-navigate-v0
    - antsoccer-arena-navigate-v0
    - cube-single-play-v0
    - cube-double-play-v0
    - scene-play-v0
    - puzzle-3x3-play-v0
    - puzzle-4x4-play-v0

- Pixel-based datasets:
    - visual-cube-single-play-v0
    - visual-cube-double-play-v0
    - visual-scene-play-v0
    - visual-puzzle-3x3-play-v0
    - visual-puzzle-4x4-play-v0

For evaluation, we report binary task success rates (percentages). Due to computational constraints, for each pixel-based environment we evaluate only the first task ("-singletask-task1").

**D4RL (Fu et al., 2020):**
To facilitate comparison with prior results, we additionally evaluate on 18 standard tasks from D4RL, including 6 antmaze tasks and 12 adroit tasks:

- D4RL Antmaze tasks:
    - antmaze-umaze-v2
    - antmaze-umaze-diverse-v2
    - antmaze-medium-play-v2

- antmaze-medium-diverse-v2
- antmaze-large-play-v2
- antmaze-large-diverse-v2

- D4RL Adroit tasks:

  - pen-human-v1, pen-cloned-v1, pen-expert-v1
  - door-human-v1, door-cloned-v1, door-expert-v1
  - hammer-human-v1, hammer-cloned-v1, hammer-expert-v1
  - relocate-human-v1, relocate-cloned-v1, relocate-expert-v1

Following the standard D4RL evaluation protocol (Fu et al., 2020), we report binary success rates for antmaze tasks and normalized returns for adroit tasks. Note that while the D4RL antmaze tasks share a similar high-level objective with OGBench navigation tasks, they differ in layout, datasets, and evaluation details.

### F.2. Hyperparameter Tuning

Complete hyperparameter lists are provided in the following tables (Tables 6, 7, and 8).

*Table 6.* **Hyperparameters for DFQL.**

| Hyperparameter | Value |
| --- | --- |
| Learning rate | 0.0003 |
| Optimizer | Adam |
| Gradient steps | 1000000 (default), 500000 (D4RL, pixel-based OGBench) |
| Minibatch size | 256 |
| MLP dimensions | [512, 512, 512, 512] |
| Nonlinearity | GELU & Tanh (additional on the last layer) |
| Target network smoothing coefficient | 0.005 |
| Discount factor $\gamma$ | 0.99 (default), 0.995 (antmaze-giant, humanoidmaze, antsoccer) |
| Image augmentation probability | 0.5 |
| Flow steps | 10 |
| Clipped double Q-learning | False (default), True (adroit, antmaze-{large, giant}-navigate) |
| coefficient $\alpha$ | Tables 7 and 8 |

*Table 7.* **Task-specific hyperparameters for offline RL.** We perform cross-task adjustments for the hyperparameter $\alpha$, which is individually tuned for each task. But in OGBench, we tune it only on the default task (denoted by (*)) and apply the resulting best hyperparameters to the other four tasks.

| Task | $\alpha$ |
|---|---|
| antmaze-large-navigate-singletask-task1-v0 (*) | 20 |
| antmaze-large-navigate-singletask-task2-v0 | 20 |
| antmaze-large-navigate-singletask-task3-v0 | 20 |
| antmaze-large-navigate-singletask-task4-v0 | 20 |
| antmaze-large-navigate-singletask-task5-v0 | 20 |
| antmaze-giant-navigate-singletask-task1-v0 (*) | 10 |
| antmaze-giant-navigate-singletask-task2-v0 | 10 |
| antmaze-giant-navigate-singletask-task3-v0 | 10 |
| antmaze-giant-navigate-singletask-task4-v0 | 10 |
| antmaze-giant-navigate-singletask-task5-v0 | 10 |
| humanoidmaze-medium-navigate-singletask-task1-v0 (*) | 10 |
| humanoidmaze-medium-navigate-singletask-task2-v0 | 10 |
| humanoidmaze-medium-navigate-singletask-task3-v0 | 10 |
| humanoidmaze-medium-navigate-singletask-task4-v0 | 10 |
| humanoidmaze-medium-navigate-singletask-task5-v0 | 10 |
| humanoidmaze-large-navigate-singletask-task1-v0 (*) | 10 |
| humanoidmaze-large-navigate-singletask-task2-v0 | 10 |
| humanoidmaze-large-navigate-singletask-task3-v0 | 10 |
| humanoidmaze-large-navigate-singletask-task4-v0 | 10 |
| humanoidmaze-large-navigate-singletask-task5-v0 | 10 |
| antsoccer-arena-navigate-singletask-task1-v0 | 10 |
| antsoccer-arena-navigate-singletask-task2-v0 | 10 |
| antsoccer-arena-navigate-singletask-task3-v0 | 10 |
| antsoccer-arena-navigate-singletask-task4-v0 (*) | 10 |
| antsoccer-arena-navigate-singletask-task5-v0 | 10 |
| cube-single-play-singletask-task1-v0 | 1 |
| cube-single-play-singletask-task2-v0 (*) | 1 |
| cube-single-play-singletask-task3-v0 | 1 |
| cube-single-play-singletask-task4-v0 | 1 |
| cube-single-play-singletask-task5-v0 | 1 |
| cube-double-play-singletask-task1-v0 | 2 |
| cube-double-play-singletask-task2-v0 (*) | 2 |
| cube-double-play-singletask-task3-v0 | 2 |
| cube-double-play-singletask-task4-v0 | 2 |
| cube-double-play-singletask-task5-v0 | 2 |
| scene-play-singletask-task1-v0 | 1 |
| scene-play-singletask-task2-v0 (*) | 1 |
| scene-play-singletask-task3-v0 | 1 |
| scene-play-singletask-task4-v0 | 1 |
| scene-play-singletask-task5-v0 | 1 |
| puzzle-3x3-play-singletask-task1-v0 | 3.3 |
| puzzle-3x3-play-singletask-task2-v0 | 3.3 |
| puzzle-3x3-play-singletask-task3-v0 | 3.3 |
| puzzle-3x3-play-singletask-task4-v0 (*) | 3.3 |
| puzzle-3x3-play-singletask-task5-v0 | 3.3 |
| puzzle-4x4-play-singletask-task1-v0 | 0.2 |
| puzzle-4x4-play-singletask-task2-v0 | 0.2 |
| puzzle-4x4-play-singletask-task3-v0 | 0.2 |
| puzzle-4x4-play-singletask-task4-v0 (*) | 0.2 |
| puzzle-4x4-play-singletask-task5-v0 | 0.2 |
| antmaze-umaze-v2 | 20 |
| antmaze-umaze-diverse-v2 | 20 |
| antmaze-medium-play-v2 | 20 |
| antmaze-medium-diverse-v2 | 20 |
| antmaze-large-play-v2 | 33 |
| antmaze-large-diverse-v2 | 33 |
| pen-human-v1 | 0.033 |
| pen-cloned-v1 | 0.033 |
| pen-expert-v1 | 0.2 |
| door-human-v1 | 0.01 |
| door-cloned-v1 | 0.01 |
| door-expert-v1 | 0.01 |
| hammer-human-v1 | 0.033 |
| hammer-cloned-v1 | 0.033 |
| hammer-expert-v1 | 0.0033 |
| relocate-human-v1 | 0.033 |
| relocate-cloned-v1 | 0.01 |
| relocate-expert-v1 | 0.01 |
| visual-cube-single-play-singletask-task1-v0 | 0.33 |
| visual-cube-double-play-singletask-task1-v0 | 0.33 |
| visual-scene-play-singletask-task1-v0 | 1 |
| visual-puzzle-3x3-play-singletask-task1-v0 | 1 |
| visual-puzzle-4x4-play-singletask-task1-v0 | 0.33 |

*Table 8.* **Task-specific hyperparameters for offline-to-online RL.** We individually tune these hyperparameters for each task.

| Task | $\alpha$ |
|---|---|
| humanoidmaze-medium-navigate-singletask-v0 | 10 |
| antsoccer-arena-navigate-singletask-v0 | 10 |
| cube-double-play-singletask-v0 | 2 |
| scene-play-singletask-v0 | 1 |
| puzzle-4x4-play-singletask-v0 | 0.2 |
| antmaze-umaze-v2 | 20 |
| antmaze-umaze-diverse-v2 | 20 |
| antmaze-medium-play-v2 | 20 |
| antmaze-medium-diverse-v2 | 20 |
| antmaze-large-play-v2 | 33 |
| antmaze-large-diverse-v2 | 33 |
| pen-cloned-v1 | 0.1 |
| door-cloned-v1 | 0.1 |
| hammer-cloned-v1 | 0.1 |
| relocate-cloned-v1 | 0.2 |

