# OpenReview forum: "Direct Flow Q-Learning"
_ICML.cc/2026/Conference — ICML 2026 regular_

### Official Review · Reviewer_5B9k · 2026-03-03

**Soundness:** 3
**Presentation:** 2
**Significance:** 2
**Originality:** 2
**Overall Recommendation:** 3
**Confidence:** 3

**Summary:**

This paper proposes Direct Flow Q-Learning (DFQL), aiming to directly train multi-step flow-matching policies in offline reinforcement learning while avoiding two existing pain points: gradient explosion/instability during end-to-end BPTT training of multi-step generation policies; and the potential loss of expressive power after distilling multi-step flows into a single-step policy.
The core approach is to inject the gradient of the terminal Q-gradient Q(s, a_K) as "a local, step-wise supervision" into the velocity field learning of each flow step, instead of backpropagating through the entire sampling chain, combined with the time weight w_k=1/(K-k) to form an alternative target.

**Compliance With Llm Reviewing Policy:**

Affirmed.

**Key Questions For Authors:**

Regarding the theoretical effective domain and error control:
Please provide an empirical curve/upper bound estimate of the deviation between the DFQL gradient and the true terminal gradient under a finite step K (non-asymptotic).


It is recommended to rerun the key baselines (especially FQL, FBRAC, and IFQL) by adding "unified codebase, unified computing budget, and unified random seed", and report wall clock and video memory usage.


Hyperparameter α sensitivity is a core limitation that you acknowledge, and we recommend adding a cross-task auto-tuning strategy (even a simple heuristic) and reporting on its stability.

**Limitations:**

yes

**Strengths And Weaknesses:**

Strengths:
The topic is well-defined and important: it proposes a direct solution to the key pain points of flow policy in offline RL (BPTT instability vs. distillation loss).

The method design is relatively clean: the terminal Q-gradient is directly injected into the velocity field, and the training process is feasible.

The experiments have broad coverage: OGBench + D4RL, including state and pixel tasks, with a large number of baselines.

Additional stability experiments and ablations (K-step, dense BC, numerical guardrail, QC combination) are provided, not just the main table.

Weaknesses:
The theoretical section reads more like a "reasoning explanation" than a rigorous unbiased guarantee: the method inherently introduces bias (ignoring per-step Jacobian chains), and consistent conclusions rely on strong assumptions and asymptotic conditions.

The hyperparameter α is significantly sensitive, and its individual tuning for each task (including finer meshes) affects the assessment of "method robustness/usability."

Some fairness and comparability issues remain: the authors reuse a large amount of FQL report results and add extra numerical protection to DFQL (although the motivation is explained), but a more systematic and unified rerun/computational reporting is still recommended.

---

> ### Author Rebuttal · Authors · 2026-03-30
>
> Thank you for the careful reading and constructive feedback. We appreciate the important concerns you raised and respond point-by-point below.
>
> **(1) Finite-K validity and theoretical scope.**
>
> Thanks for asking this valid question. We agree that DFQL is not a strictly unbiased finite-K gradient estimator. Its key approximation is to drop the inter-step Jacobian chain to avoid BPTT instability for multi-step flow policies; this introduces bias by design. More precisely, Appendix A.2 does not prove consistency of the DFQL gradient itself with the true terminal gradient. Rather, it establishes that the resulting update direction induced by DFQL asymptotically aligns with the terminal guidance direction. We will formalize this distinction in the final version to more accurately reflect the theoretical properties of DFQL.
>
> To address the non-asymptotic concern, we added an empirical finite-K analysis. For one virtual SGD step on the actor objective, we measure the induced terminal-action change $\Delta a_K$ and compare it with the terminal guidance direction using cosine similarity and the fraction of samples with positive inner product (“positive fraction”). The results for $K \in \\{1, 4, 10, 20 \\} $ across three representative tasks are summarized in the table below:
>
> | Task | $K=1$ | $K=4$ | $K=10$ | $K=20$ |
> | :--- | :--- | :--- | :--- | :--- |
> | Humanoidmaze-medium-navigate | 0.714 / 96.87% | 0.731 / 100% | 0.774 / 100% | 0.830 / 100% |
> | Cube-double-play | 0.936 / 100% | 0.956 / 100% | 0.961 / 100% | 0.971 / 100% |
> | Scene-play | 0.750 / 93.75% | 0.833 / 96.87% | 0.859 / 100% | 0.942 / 100% |
>
> The positive fraction indicates how often the update has positive inner product with the desired guidance direction, which under standard optimization intuition corresponds to a locally effective update. Thus, although humanoidmaze only reaches cosine 0.83 at K=20, its positive fraction is already 100% from K=4 onward, meaning all sampled updates remain directionally reliable. Overall, these results support our interpretation that DFQL is a biased but stable surrogate objective: even at finite K, its induced terminal updates remain positively aligned with the desired guidance direction.
>
> **(2) Sensitivity to 𝛼.**
> Thank you for the constructive feedback on 𝛼-sensitivity. To investigate this, we evaluated an adaptive 𝛼 strategy that dynamically rebalances the BC and Q signals. Our testing shows that while such a mechanism offers some benefits, it introduces additional complexity regarding the adaptive scale and the schedule's own hyperparameters. Due to the character limit and to avoid redundancy, we provide detailed experimental results and further discussion in our response to Reviewer n12u (3). We view adaptive 𝛼 as a promising future direction, though it is not yet mature enough to be integrated into the current framework.
>
> **(3) Comparability and computational reporting.**
>
> Thanks for asking this valid question. DFQL is an incremental modification built on the same FQL codebase, with aligned backbone and training protocol. Before reusing the FQL-reported baselines, we fully reproduced the official FQL code on the 50 OGBench tasks. While some individual task scores vary across seeds or machines, the final mean matches the reported value after rounding (44), indicating that directly using the FQL baseline does not change the overall comparison. We will add this reproduction result in the final version.
>
> In addition, under the reviewer-requested unified setting, we reran the key baselines and profiled both runtime and memory usage. In terms of inference time, DFQL is lower than IFQL and close to FBRAC, although still slower than FQL. In terms of training time, DFQL has the highest per-update training cost among these methods. However, a higher per-update cost does not necessarily imply slower progress toward a target performance level: on the three representative tasks used in our ablations, DFQL reaches the best average performance achieved by FQL in fewer training stages, while also achieving better average performance in later training. Due to the character limit and to avoid redundancy, we provide more detailed results and discussion in our response to Reviewer PSwU (3). We will add runtime comparison to the ablation section in the final version.
>
> For memory, under JAX on-demand allocation, the GPU memory-allocation peak within 100K training steps is about 522.5/502.7/508.4/493.5 MiB for DFQL/FQL/FBRAC/IFQL, respectively. Due to the limited rebuttal time, profiling in this mode substantially slows training, so we only conducted a brief measurement over the first 100K steps rather than over the full 1M-step training run. DFQL has the highest memory usage among them, consistent with its more complex computation graph. However, since the overall differences are small, we believe the different policy parameterizations do not differ substantially in memory usage.

---

> > ### Author Rebuttal · Reviewer_5B9k · 2026-04-03
> >
> > Thank you for the rebuttal. I have no further questions and will carefully consider the final score

---

### Official Review · Reviewer_n12u · 2026-03-11

**Soundness:** 4
**Presentation:** 4
**Significance:** 3
**Originality:** 3
**Overall Recommendation:** 5
**Confidence:** 4

**Summary:**

This paper proposes Direct Flow Q-Learning (DFQL), a framework that stabilizes the optimization of multi-step Flow Matching policies in offline RL. Instead of relying on unstable Backpropagation Through Time (BPTT) or capacity-limiting policy distillation, DFQL injects terminal Q-value gradients directly into the velocity field at each generation step via a temporally weighted surrogate objective. The method achieves state-of-the-art results across 73 continuous control tasks in OGBench and D4RL, demonstrating both theoretical soundness and empirical effectiveness.

**Compliance With Llm Reviewing Policy:**

Affirmed.

**Final Justification:**

My concerns have been fully addressed.

**Key Questions For Authors:**

1. Will the codebase be open-sourced upon publication to facilitate reproducibility, given the complexity of the dense BC and temporal weighting implementations?

2. Could the authors provide a brief profiling of the wall-clock training and inference times for DFQL vs. FQL (or other one-step distillation baselines) on a representative task?

**Limitations:**

yes

**Strengths And Weaknesses:**

**Strengths:**
- Sound Theoretical Foundation. The motivation is crystal clear. Furthermore, the derivation of asymptotic consistency (Appendix A.2) rigorously justifies the approach. Although DFQL introduces biased approximations to bypass BPTT instability (e.g., state-sensitivity pruning), the temporal weighting schedule $w_k = 1/(K-k)$ elegantly guarantees that the dominant gradient direction aligns with the true objective when $\Delta t \to 0$. This represents a solid theoretical contribution.

- Extensive and Fair Empirical Evaluation. The evaluation spans 73 diverse tasks. Notably, the authors demonstrate high scientific integrity by ensuring baseline fairness (e.g., removing the bounded tanh activation for FQL in Table 3 to avoid penalizing the baseline).

- The seamless integration of DFQL with Q-Chunking (QC-DFQL) highlights the framework's potential as a robust backbone for other orthogonal RL algorithmic enhancements.


**Weaknesses:**
- As acknowledged in the Limitations, the framework relies heavily on per-task tuning of the coefficient $\alpha$ (ranging from 0.0033 to 33, per Tables 7 and 8). This task-specific sensitivity somewhat diminishes the out-of-the-box applicability of the method.

- The paper claims to preserve the expressive capacity of multi-step generation, which inherently incurs higher inference costs than single-step distillation methods. The experimental section would be much stronger with a quantitative wall-clock inference time comparison.

- Minor Comments: In Section 3.3, Equation 20, the expectation is taken over $a_0 \sim \mathcal{N}(0, I_d)$, and the regression target is formulated as $(a - a_0)$. However, the text immediately below defines the interpolated particle using $a_0^{BC}$. To maintain strict notational consistency and avoid any confusion with the initial noise of the standard RL trajectory, the $a_0$ inside Equation 20 should be corrected to $a_0^{BC}$.

---

> ### Author Rebuttal · Authors · 2026-03-30
>
> Thank you for the thoughtful and positive review, and for recognizing the theoretical clarity, empirical breadth, and careful baseline treatment of our work. We also appreciate your constructive suggestions on reproducibility, efficiency profiling, and practical usability, and respond point-by-point below.
>
> **(1) Open-sourcing and reproducibility.**
>
> Yes. We are committed to open-sourcing the code upon publication. We will retain the current OpenReview submission materials and prepare a cleaner public repository with clearer reproduction instructions and tuning guidance, and include the repository link in the final version.
>
> **(2) Wall-clock training/inference time.**
>
> Thank you for highlighting the importance of the efficiency-performance tradeoff. To provide a clearer comparison, we have profiled the runtime for all flow-based baselines. In terms of inference time, DFQL is faster than IFQL and comparable to FBRAC, though slower than the one-step FQL. In terms of training time, DFQL has a higher per-update training cost, but this does not necessarily translate to slower overall progress. On the three representative tasks in our ablation study, DFQL reaches FQL’s peak average performance in fewer training stages while achieving superior final results. Due to the character limit and to avoid redundancy, we provide the full quantitative results (wall-clock time, and performance-over-time metrics) in our response to Reviewer PSwU (3). We will incorporate this comprehensive runtime comparison into the final version's ablation section.
>
> **(3) 𝛼-sensitivity and practical usage.**
>
> Thank you for the constructive feedback regarding the sensitivity of 𝛼 and its implications for out-of-the-box usability. To address this, we will add a clearer practical tuning note in the appendix of the final version, so that users can apply the method more easily. Based on our experience, tuning 𝛼 is better viewed as a two-level process: first, a coarse-grained search over the overall scale, to determine the rough order of magnitude of 𝛼; second, a small-range refinement within that scale.
> Further, for the second-stage local refinement, we additionally tested an adaptive-𝛼 method proposed in recent work (One-Step Generative Policies with Q-Learning: A Reformulation of MeanFlow, AAAI 2026), as an automatic fine-tuning tool. Its basic idea is to monitor the BC loss relative to its running average and dynamically adjust 𝛼 accordingly, so as to rebalance the BC and Q signals during training. More concretely, when the BC loss suddenly becomes larger, it reduces the Q weight; when the BC loss suddenly becomes smaller, it increases the Q weight, so as to avoid overly abrupt instability in the imitation term.
> Under this setup, we first choose a coarse initial scale: 𝛼 =10 for humanoidmaze-medium-navigate-singletask-v0, and 𝛼 =1 for cube-double-play-singletask-v0 and scene-play-singletask-v0. We then apply adaptive-𝛼 for local refinement around that scale. The resulting performance compared with fixed 𝛼 is summarized below:
>
> | Task | Fixed 𝛼 | Adaptive 𝛼 |
> | :--- | :---: | :---: |
> | Humanoidmaze-medium-navigate | 35 | **38 ± 24** |
> | Cube-double-play | 46 | **48 ± 16** |
> | Scene-play | **97** | 93 ± 7 |
>
> Compared with fixed 𝛼, this yields small improvements on cube and humanoidmaze, but a slight drop on scene. We view this as evidence that the method can indeed provide some practical assistance by reducing part of the manual fine-tuning effort. For example, on tasks such as cube-double, one may start directly from 𝛼=1 and let the adaptive process refine it, rather than manually tuning the initial value further.
>
> At the same time, we also found some issues with this method: it does not truly remove tuning, but instead shifts part of the burden to the choice of the adaptive schedule itself. In DFQL, we found that the original setting of this method produces almost no noticeable dynamics, so we switched to a more easily triggered and milder version, using a trigger region around 1.1/0.9 and multiplicative updates of ×1.1 or ×0.9 after triggering (here, the “trigger region” refers to how far the current BC loss deviates from its running average before an automatic update of 𝛼 is applied). Overall, this means that while the method can reduce part of the small-range tuning effort, it still requires task or architecture-dependent adjustment of the trigger sensitivity and update magnitude.
> Therefore, we treat this adaptive-alpha method as an optional plugin for users. As it is outside the main scope of the present paper, and we do not develop new theoretical analysis or DFQL-specific improvement for it here, we do not treat it as part of the paper’s method. Instead, we plan to include this implementation in a later version of the released codebase, so that practitioners can try it more easily.
>
> **(4) Minor notation issue in Eq. 20.**
>
> Thank you for catching this. We will fix it in the final version.

---

> > ### Author Rebuttal · Reviewer_n12u · 2026-04-02
> >
> > Thank you for the rebuttal. I think your response has resolved my fundamental issue. I keep my original score.

---

### Official Review · Reviewer_PSwU · 2026-03-12

**Soundness:** 3
**Presentation:** 3
**Significance:** 3
**Originality:** 3
**Overall Recommendation:** 4
**Confidence:** 4

**Summary:**

The submission introduces a new method for learning actors via iterative computation. Indeed, as opposed to traditional methods, which rely on a single-step prediction, this new method produces an action that is the result of multiple passes via a neural network. The presented method relies on a simplified version of backpropagation through time, which does not seem to suffer from exploding gradients or biased estimation due to the approximation.

**Compliance With Llm Reviewing Policy:**

Affirmed.

**Final Justification:**

As explained in the Rebuttal Acknowledgement, the authors adressed my concerns and promised to include the newly added details in the final version of their work.

Therefore, I recommend accepting this submission.

**Key Questions For Authors:**

N/A

**Limitations:**

Please refer to the weaknesses.

**Strengths And Weaknesses:**

Strengths:

A. The presented algorithm is clearly explained and motivated.

B. The presented algorithm seems effective compared to related approaches on a wide variety of tasks.

Weaknesses:

1. As opposed to FQL, which only provides benefits when a dataset is available, the presented method could also provide a benefit in an online scenario, without prior data. Evaluating the proposed approach in this setting would be beneficial to fully understand the scope of the contribution.

2. While the method is motivated from a flow matching perspective, the simplifications to make the algorithm work steer the presented method from its initial motivation. Providing an explanation of the relation between flow matching and the proposed approach would be valuable.

3. While the limitations are discussed, a quantitative comparison focusing on floating point operations, training time, and inference time between DFQL and FQL is needed to fully grasp the tradeoff between increased performance and additional computational budget.

4. Multiple anonymized references submitted to ICLR are used across the submission. As the review process of ICLR is now terminated, those works can be de-anonymized.

---

> ### Author Rebuttal · Authors · 2026-03-30
>
> Thank you for the thoughtful and positive review, and for recognizing the clarity of our presentation and the broad empirical effectiveness of DFQL. We appreciate these constructive suggestions and respond point-by-point below.
>
> **(1) Online setting without prior data.**
>
> Thank you for this constructive observation. We agree that a theoretical potential of DFQL is its independence from a pre-existing dataset, which could allow it to function in pure online “from-scratch” scenarios. In this work, we intentionally focus on the offline and offline-to-online regimes because these settings typically involve learning from diverse datasets with highly multimodal action distributions, where flow-matching policies are uniquely suited due to their exceptional expressive capacity in modeling complex behaviors. Your perspective has inspired our next stage of research, as evaluating DFQL in pure online settings is a valuable direction to fully map out the method’s potential, and we will discuss this exciting avenue for future work in the final version of the paper.
>
> **(2) Relation between flow matching and DFQL.**
>
> Thanks for asking this valid question. The relationship between DFQL and Flow Matching (FM) centers on the principle of local supervision in velocity space. While DFQL is inspired by Conditional Flow Matching (CFM), a direct RL analogue is non-trivial because RL typically lacks the explicit target action distributions required for standard FM training. Consequently, DFQL is designed as a numerically stable surrogate rather than a vanilla FM objective. We retain the flow-matching parameterization, probability path, and iterative structure, but introduce an RL-specific surrogate to inject terminal signals into each step. Techniques such as direct guidance injection, state-sensitivity pruning, and temporal weighting are specifically implemented to ensure optimization stability when supervision is only available at the terminal action. We will update the final version to formalize these distinctions and the underlying FM-based motivation.
>
> **(3) Computational cost: FLOPs/training/inference time.**
>
> Thank you for highlighting the importance of the efficiency-performance tradeoff. To provide an explicit comparison, we profiled all major flow-based baselines under a unified setup (same machine, codebase, and 8 seeds) in a representative environment (cube-double-play-singletask-v0, consistent with the representative task selection in FQL), as summarized in the table below:
>
> | Method | Inference (ms) | Training (ms) |
> | :--- | :---: | :---: |
> | FQL | 0.2267 | 2.5589 |
> | DFQL (Ours) | 0.4820 | 3.9323 |
> | FBRAC | 0.4932 | 3.4652 |
> | IFQL | 0.9267 | 1.9791 |
>
> In terms of inference time, DFQL’s inference speed is slower than one-step FQL, roughly on par with FBRAC, and faster than IFQL. This is consistent with their structures: FQL is one-step, FBRAC uses a similar iterative inference process, and IFQL requires selecting among N candidates. For training, DFQL involves a higher training cost per optimization step, this is primarily because, like FBRAC, it propagates RL supervision to every iterative step. A secondary reason is the extra dense-BC supervision, which makes DFQL slightly slower than FBRAC.
>
> However, DFQL is not necessarily disadvantaged when measured by the training process required to reach the same target performance within a total training budget of 1M steps. On three representative tasks (cube-double-play-singletask-v0, scene-play-singletask-v0, and humanoidmaze-medium-navigate-singletask-v0), we define the target threshold as the best FQL average return (38, 83, and 32, respectively) using 8-seed averages and evaluations every 100K steps. DFQL exceeds these thresholds and never drops below them again at 700K, 100K, and 700K steps, respectively (average 500K, latest 700K). Therefore, although DFQL is more expensive per update, the training duration it requires to reach the same performance level is not necessarily worse, while its final average performance is higher.
>
> For FLOPs, we understand the value of a more theory-oriented complexity measure. Since these methods differ not only in forward passes but also in iterative generation and candidate selection, we prioritized wall-clock profiling in rebuttal as the more direct practical cost measure, and we will expand the discussion of computational cost in the final version. We will also add a dedicated comparison of training and inference speed in the ablation sections.
>
> **(4) De-anonymizing ICLR references.**
>
> Thank you for pointing this out. We will de-anonymize those references in the final version now that the ICLR review process has concluded.

---

> > ### Author Rebuttal · Reviewer_PSwU · 2026-04-01
> >
> > The authors adressed the concerns and promised to include the newly added details in the final version of their work.

---

### Decision · Program_Chairs · 2026-04-30

**Decision:**

Accept (regular)

**Comment:**

All reviewers agree the paper studies a relevant and well-motivated problem, a that the proposed method is sound, derived from theoretical insights, and has stronger empirical performance agains relevant baselines. The main concerns were about the sensitivity to the hyperparameter alpha and that the theory is not fully rigorous, but rather contains some heuristic approximations. I think the authors nicely addressed both points in their rebuttal with both intuitive explanations and additional experiments. I thus recommend acceptance and encourage the authors to revise the manuscript accordingly.